# UNSUPERVISED OBJECT LEARNING VIA COMMON FATE

## ABSTRACT

Learning generative object models from unlabelled videos is a long standing problem and required for causal scene modeling. We decompose this problem into three easier subtasks, and provide candidate solutions for each of them. Inspired by the Common Fate Principle of Gestalt Psychology, we first extract (noisy) masks of moving objects via unsupervised motion segmentation. Second, generative models are trained on the masks of the background and the moving objects, respectively. Third, background and foreground models are combined in a conditional "dead leaves" scene model to sample novel scene configurations where occlusions and depth layering arise naturally. To evaluate the individual stages, we introduce the FISHBOWL dataset positioned between complex real-world scenes and common object-centric benchmarks of simplistic objects. We show that our approach allows learning generative models that generalize beyond the occlusions present in the input videos, and represent scenes in a modular fashion that allows sampling plausible scenes outside the training distribution by permitting, for instance, object numbers or densities not observed in the training set.

## 1 INTRODUCTION

Machine learning excels if sufficient training data is available that is representative of the task at hand. In recent years, this *i.i.d.* data paradigm has been shown not only to apply for pattern recognition problems, but also for generative modeling (Goodfellow et al., 2014). In practice, the amount of data required to reach a given level of performance will depend on the dimensionality of the data. The generation of high-dimensional images thus either requires huge amounts of data (Karras et al., 2020) or clever methods that exploit prior information, for instance on multi-scale structure or compositionality (Razavi et al., 2019).

Imagine we would like to automatically generate realistic images of yearbook group photos. A "brute force" approach would be to collect a massive dataset and train a large GAN (Goodfellow et al., 2014), hoping that the model will not only learn typical backgrounds, but also the shape of individual humans (or human faces), and arrange them into a group. A more modular approach, in contrast, would be to learn *object models* (e.g., for faces, or humans), and learn in which positions and arrangements they appear, as well as typical backgrounds. This approach would be more data-efficient: each training image would contain multiple humans, and we would thus effectively have more data for the object learning task. In addition, the sub-task would be lower-dimensional than the original task. Finally, if we leave the i.i.d. setting (by, say, having a second task with different group sizes), the modular approach would lend itself more readily to knowledge transfer.

Object-centric approaches aim to capture this compositionality and have been considerably improved over the past few years (e.g., Locatello et al. 2020; Engelcke et al. 2021). However, these models tend to be difficult to train and do not scale well to visually more complex scenes. In addition, the commonly employed end-to-end learning approaches make it difficult to dissect the causes of these difficulties and what principles may be crucial to facilitate unsupervised object learning.

In human vision, the *Principle of Common Fate* of Gestalt Psychology (Wertheimer, 2012) has been shown to play an important role for object learning (Spelke, 1990). It posits that elements that are moving together tend to be perceived as one—a perceptual bias that may have evolved to be able to recognize camouflaged predators (Troscianko et al., 2009).

In our work, we show that this principle can be successfully used also for machine vision by using it in a multi-stage object learning approach (Fig. 1): First, we use unsupervised motion segmentation to obtain a candidate segmentation of a video frame. Second, we train generative object and background models on this segmentation. While the regions obtained by the motion segmentation

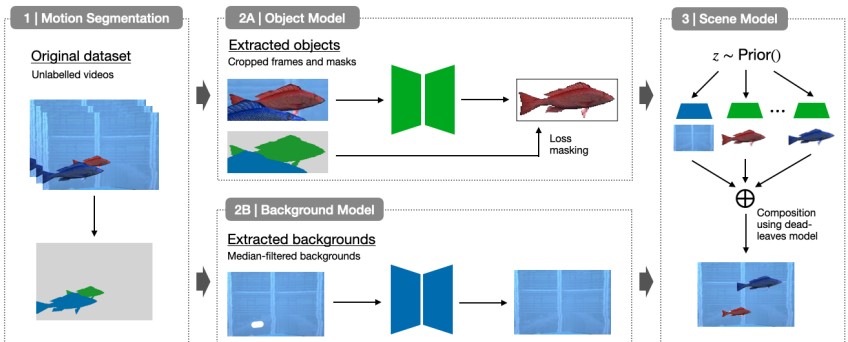

Figure 1: We propose a multi-stage approach for learning object-centric generative scene models: (1) Motion segmentation detects moving objects in the input videos. The predicted (noisy) segmentation masks are used to (2A) extract object crops for training a generative object model and (2B) extract backgrounds for training a generative background model. (3) A scene model combines the object and background models to sample novel scenes, permitting interventions on variables such as the background and the number and locations of fish.

are caused by objects moving in 3D, only *visible* parts can be segmented. To learn the actual objects (i.e., the causes), a crucial task for the object model is learning to generalize beyond the occlusions present in its input data. To measure success, we provide a dataset including object ground truth. As the last stage, we show that the learned object and background models can be combined into a flexible scene model that allows *sampling manipulated* novel scenes. Thus, in contrast to existing object-centric models trained end-to-end, our work aims at decomposing object learning into evaluable subproblems and testing the potential of exploiting object motions for building scalable object-centric models that allow for causally meaningful interventions in generation.

Summing up, the present work makes the following contributions:

- We provide the novel FISHBOWL dataset, positioned between simplistic toy scenarios and real world data, providing ground truth information for evaluating causal scene models.
- We show that the *Common Fate Principle* can be succesfully used for object learning by proposing a multi-stage object learning approach based on this principle.
- We demonstrate that the generative object and background models learned in this way can be combined into flexible scene models allowing for controlled out-of-distribution sampling.

The dataset with rendering code and models including training code will be made publicly available.

## 2    RELATED WORK

**Modular scene modeling.**   The idea to individually represent objects in a scene is not new. One approach, motivated by the analysis-by-synthesis paradigm from cognitive science (Bever & Poeppel, 2010), assumes a detailed specification of the generative process and infers a scene representation by trying to invert this process (Kulkarni et al., 2015; Wu et al., 2017; Jampani et al., 2015). Many methods instead aim to also learn the generative process in an unsupervised way, see Greff et al. (2020) for a recent survey. Several models use a recurrent approach to sequentially decompose a given scene into objects (Eslami et al., 2016; Stelzner et al., 2019; Kosiorek et al., 2018; Gregor et al., 2015; Mnih et al., 2014; Yuan et al., 2019; Engelcke et al., 2020; Weis et al., 2020; von Kügelgen et al., 2020; Burgess et al., 2019), or directly learn a partial ordering (Heess et al., 2011; Le Roux et al., 2011). This sequential approach has also been extended with spatially-parallel components in (Dittadi & Winther, 2019; Jiang et al., 2019; Lin et al., 2020b; Chen et al., 2020). Other methods infer all object representations in parallel (Greff et al., 2017; van Steenkiste et al., 2018), with subsequent iterative refinement (Greff et al., 2019; Veerapaneni et al., 2019; Locatello et al., 2020; Nanbo et al., 2020). Whereas most of the above models are trained using a reconstruction objective—usually in a variational framework (Kingma & Welling, 2014; Rezende et al., 2014)—several works have also extended GANs (Goodfellow et al., 2014) to generate scenes in a modular way (Yang et al., 2017; Turkoglu et al., 2019; Nguyen-Phuoc et al., 2020; Ehrhardt et al., 2020; Niemeyer & Geiger, 2021). Those approaches typically use additional supervision such as ground-truth segmentation or additional views, with Ehrhardt et al. (2020); Niemeyer & Geiger (2021) being notable exceptions. While most methods can decompose a given scene into its constituent objects, only few are fully-generative in the sense that they can generate novel scenes (Lin et al., 2020a; Ehrhardt et al., 2020; Engelcke et al., 2020; von Kügelgen et al., 2020; Engelcke et al., 2021; Niemeyer & Geiger, 2021; Dittadi & Winther, 2019).

Figure 2: The FISHBOWL dataset: each video contains 128 frames with ground truth segmentation; renderings and masks (without occlusion) of every fish and background are provided in the validation and test sets.

Our approach differs from previous works in the following three key aspects. First, previous approaches typically train a full scene model in an end-to-end fashion and include architectural biases that lead to the models decomposing scenes into objects. While elegant in principle, those methods have not be shown to scale to more realistic datasets yet. Using a multi-stage approach, as in the present work, enables re-use of existing computer vision methods (such as unsupervised motion segmentation) for well-studied sub-tasks and therefore scales more easily to visually complex scenes. Second, while some existing methods make use of temporal information from videos (Lin et al., 2020a; Crawford & Pineau, 2020; Kosiorek et al., 2018; Weis et al., 2020), they do not explicitly use motion signals to discover (i.e., segment) objects. Inspired by the development of the human visual system (Spelke, 1990), we instead explicitly include this segmentation cue in our approach. Third, most existing fully-generative approaches use a spatial mixture model to compose objects into a scene (Ehrhardt et al., 2020; Engelcke et al., 2020; 2021). While this simplifies training, it clearly does not match the true, underlying scene generation process. In this work, we instead follow the dead leaves model-approach of von Kügelgen et al. (2020) and scale it to more complex scenes.

**Motion Segmentation.** We require an unsupervised motion segmentation method that is able to segment multiple object instances. For this, we build on a line of work that tracks points with optical flow, and then performs clustering in the space of the resulting point trajectories (Brox & Malik, 2010; Ochs et al., 2014; Ochs & Brox, 2012; Keuper et al., 2015). Motion segmentation methods that require supervision (Dave et al., 2019; Xie et al., 2019; Tokmakov et al., 2017a;b) or only perform binary motion segmentation (Yang et al., 2021; 2019; Ranjan et al., 2019) are not applicable in our unsupervised setting.

**Learning from motion.** In the present work, we propose exploiting motion information to decompose a scene into objects, and to learn generative object and scene models. Motion information is believed to be an important cue for the development of the human visual system (Spelke, 1990) and has also been used as a training signal for computer vision (Pathak et al., 2017; Dorfman et al., 2013; Mahendran et al., 2018a;b). While similar in spirit, these works however do not address learning of generative object and scene models.

## 3  THE FISHBOWL DATASET

Several video datasets have been used for object-centric representation learning before (Weis et al., 2020; Yi et al., 2019; Ehrhardt et al., 2020; Kosiorek et al., 2018). The ground truth object masks and appearances provided by those datasets, however, only cover the visible parts of the scene. In order to evaluate the capabilities of the object model to infer and represent the full objects even in the presence of occlusions, we propose the novel FISHBOWL dataset positioned between complex real world data and simplistic toy datasets. This dataset consist of $20,000$ training and $1,000$ validation and test videos recorded from a publicly available WebGL demo of an aquarium,[1] each with a resolution of $480 \times 320$px and 128 frames. We adapted the rendering to obtain ground truth segmentations of the scene and the ground truth unoccluded background and objects (Fig. 2). More details regarding the recording setup can be found in the supplement.

## 4  A MULTI-STAGE APPROACH FOR UNSUPERVISED SCENE MODELLING

We model an image $\mathbf{x}$ as a composition of a background ($\mathbf{m}_0 = \mathbb{1}, \mathbf{x}_0$) and an ordered list of objects $(\mathbf{m}_i, \mathbf{x}_i)$, each represented as a binary mask $\mathbf{m}_i$ and appearance $\mathbf{x}_i$. The background and objects are composed into a scene using a simple "dead leaves" model, i.e., the value of each pixel $(u, v)$ is determined by the foremost object covering that pixel. We propose a multi-stage approach (Fig. 1) for learning generative models representing objects, backgrounds and scenes in this fashion.

---

[1]http://webglsamples.org/aquarium/aquarium.html, 3-clause BSD license

STAGE 1: MOTION SEGMENTATION—OBTAINING CANDIDATE OBJECTS FROM VIDEOS

As a first step, we use unsupervised motion segmentation to obtain candidate segmentations of the input videos. We build on the minimum cost multicut method by Keuper et al. (2015), which tracks a subset of the pixels through the video using optical flow and then, inspired by the *Common Fate Principle* mentioned earlier, clusters the trajectories based on pairwise motion affinities. We use the original implementation of the authors, but replace the postprocessing required to obtain a dense segmentation with a simpler but faster non-parametric watershed algorithm (Beucher & Meyer, 1993) followed by computing spatiotemporal connected components (Silversmith, 2021).

The quality of the motion segmentation critically depends on the quality on the optical flow estimation, so we explore different models for that step. ARFlow (Liu et al., 2020) is the current state of the art self-supervised optical flow method that combines a common warping based objective with self-supervision using various augmentations. We use the published pretrained models as well as a variant trained on the Fishbowl dataset (see supplement for details). Similar augmentations as used by ARFlow can alternatively be used to synthesize training data for supervised methods, as done for generating the FlyingChairs and FlyingThings datasets (Dosovitskiy et al., 2015; Mayer et al., 2016). We experiment with FlowNet 2.0 (Ilg et al., 2017) and the more recent RAFT (Teed & Deng, 2020) trained on those two datasets.

To obtain background masks for training the background model, it is not necessary to differentiate between multiple object instances. We aim for a low rate of foreground pixels being mistaken for background pixels, while background pixels mistaken for foreground are of less concern. Hence, we use an ensemble of different background-foreground segmentation models from the bgslibrary (Sobral, 2013). Based on early experiments, we used the PAWKS (St-Charles et al., 2016), LOBSTER (St-Charles & Bilodeau, 2014), $\Sigma - \Delta$ estimation (Manzanera & Richefeu, 2007) and static frame differences and label every pixel detected as foreground by either of the methods as a foreground pixel. We found that this rather simple model can faithfully remove the foreground objects in most of the cases. We provide additional details in the appendix.

STAGE 2A: OBJECT MODEL—LEARNING TO GENERATE UNOCCLUDED, MASKED OBJECTS

**Object extraction.** We use the bounding boxes of the candidate segmentation to extract object crops from the original videos and rescale them to a common size of $128 \times 64$px. We filter out degenerate masks by ignoring all masks with an area smaller than 64 pixels and only considering bounding boxes with a minimum distance of 16px to the frame boundary. Accordingly, we extract the candidate *segmentation masks* $\mathbf{m}_0, \ldots, \mathbf{m}_K$ for each crop. For notational convenience, we take $\mathbf{m}_0$ and $\mathbf{m}_1$ to correspond to the background and the object of interest (i.e., that whose bounding box was used to create the crop), respectively, so that $\mathbf{m}_k$ with $k \geq 2$ correspond to the masks of other objects.

**Task.** We use the segmented object crops for training a $\beta$-VAE-based generative object model (Higgins et al., 2017). Input to the model is the object crop without the segmentation, output is the reconstructed object appearance including the binary object mask. We train the model with the standard $\beta$-VAE loss with an adapted reconstruction term including both the appearance and the mask. For an input batch, let $\mathbf{c}$ and $\mathbf{m}_{0:K}$ be the ground truth crops with candidate segmentations, and $\hat{\mathbf{c}}$ and $\hat{\mathbf{m}}$ the reconstructed object appearances (RGB values for each pixel) and shapes (foreground probabiliy for each each pixel). The reconstruction loss $\mathscr{L}_R$ for these objects is then the weighted sum of the pixel-wise MSE for the appearance and the pixel-wise binary cross entropy for the mask:

$$\mathscr{L}_{R,\text{appear.}} = \sum_i \left( \sum_{u,v} \mathbf{m}_1^{(i)}(u,v) \left\| \mathbf{c}^{(i)}(u,v) - \hat{\mathbf{c}}^{(i)}(u,v) \right\|_2^2 / \sum_{u,v} \mathbf{m}_1^{(i)}(u,v) \right),$$

$$\mathscr{L}_{R,\text{mask}} = \sum_i \left( \sum_{u,v} [\mathbf{m}_0^{(i)} + \mathbf{m}_1^{(i)}](u,v) \cdot \text{BCE}\big[\mathbf{m}_1^{(i)}(u,v), \hat{\mathbf{m}}^{(i)}(u,v)\big] / \sum_{u,v} [\mathbf{m}_0^{(i)} + \mathbf{m}_1^{(i)}](u,v) \right).$$

As the task for the object model is to only represent the central object in each crop, we restrict the appearance loss to the candidate mask of the object ($\mathbf{m}_1$) and the mask loss to the union of the candidates masks of the object and the background ($\mathbf{m}_0 + \mathbf{m}_1$). Importantly, the reconstruction loss is not evaluated for pixels belonging to other objects according to the candidate masks. Therefore, the object model is not penalized for completing object parts that are occluded by another object.

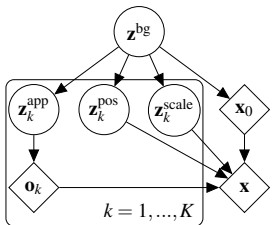

Figure 3: Causal graph for our scene model; circles and diamonds denote random and deterministic quantities.

Table 1: Segmentation performance of the unsupervised motion segmentation (Keuper et al., 2015) for different optical flow estimators.

| | Optical flow | IoU | | Recall | |
|---|---|---|---|---|---|
| Estimator | Training data | Background | Objects | @0.0 | @0.5 |
| ARFlow | KITTI | 0.874 | 0.246 | 0.828 | 0.199 |
| ARFlow | Sintel | 0.890 | 0.243 | 0.809 | 0.213 |
| ARFlow | Fishbowl | 0.873 | 0.248 | 0.842 | 0.204 |
| RAFT | FlyingThings | 0.930 | 0.318 | 0.663 | 0.351 |
| FlowNet 2 | FlyingThings | **0.934** | **0.365** | **0.674** | **0.416** |

**Learning object completion via artificial occlusions.** To encourage the model to correctly complete partial objects, we use artificial occlusions as an augmentation during training. Similar to a denoising autoencoder (Vincent et al., 2008), we compute the reconstruction loss using the unaugmented object crop. We consider two types of artificial occlusions: first, we use a *cutout* augmentation (DeVries & Taylor, 2017) placing a variable number of grey rectangles on the input image. As an alternative, we use the candidate segmentation to place another, randomly shifted object from the same input batch onto each crop.

**Model.** We use a $\beta$-VAE with 128 latent dimensions. The encoder is a ten layer CNN, the appearance decoder is a corresponding CNN using transposed convolutions (Dumoulin & Visin, 2018) and one additional convolutional decoding layer. We use a second decoder with the same architecture but only a single output channel do decode the object masks. During each epoch, we use crops from two random frames from every object. We train our model for 60 epochs using Adam (Kingma & Ba, 2015) with a learning rate of $10^{-4}$, which we decrease by a factor of 10 after 40 epochs. We chose the optimal hyperparameters for this architecture using grid searches. More details regarding the model architecture and the hyperparameters are provided in the supplement.

STAGE 2B: BACKGROUND MODEL—LEARNING TO GENERATE UNOCCLUDED BACKGROUNDS

**Task.** We use an ensemble of background extraction techniques outlined above to estimate background scenes for each frame. We train a $\beta$-VAE on these backgrounds using the appearance loss $\mathcal{L}_{R,\text{appear.}}$ with the inferred background mask, without any additional cutout or object augmentation.

**Architecture.** The $\beta$-VAE has the same architecture as the object model, but only uses a single decoder for the background appearance. We do not focus on a detailed reconstruction of background samples and limit the resolution to $96 \times 64$px. When sampling scenes, the outputs are upsampled to the original resolution of $480 \times 320$px using bilinear interpolation.

STAGE 3: SCENE MODEL—LEARNING TO GENERATE COHERENT SCENES

In the final stage, we combine the object and background model into a scene model that allows sampling novel scenes. As the scene model can reuse the decoders from the previous stages, its main task is to model the parameters defining the scene composition such as object counts, locations and dependencies between the background and the object latents. Compared to an end-to-end approach, the complexity of the learning problem is greatly reduced in this setting. It is straightforward to generalize the scene model beyond the training distribution: E.g., it is easy to sample more objects than observed in the input scenes.

We use a scene model following the causal graph depicted in Fig. 3: First, we sample a background latent $\mathbf{z}^{\text{bg}}$ which describes global properties of the scene such as its composition and illumination; $\mathbf{z}^{\text{bg}}$ is then decoded by the background model into a background image $\mathbf{x}_0$. Conditioned on the background latent, we sequentially sample $K$ tuples $(\mathbf{z}_k^{\text{app}}, \mathbf{z}_k^{\text{pos}}, \mathbf{z}_k^{\text{scale}})$ of latents encoding appearance, position, and scale of object $k$, respectively; the number of objects $K$ is sampled conditional on $\mathbf{z}^{\text{bg}}$ as well. Each appearance latent $\mathbf{z}_k^{\text{app}}$ is decoded by the object model into a masked object $\mathbf{o}_k = (\mathbf{m}_i, \mathbf{x}_i)$, which is subsequently re-scaled by $\mathbf{z}_k^{\text{scale}}$ and placed in the scene at position $\mathbf{z}_k^{\text{pos}}$ according to a dead-leaves model (i.e., occluding previously visible pixels at the same location).

Due to the formulation of the model, we are flexible in specifying the conditional and prior distributions needed to generate samples. A particular simple special case is to sample all latents (indicated as circles in Fig. 3) independently. This can be done by informed prior distributions, or by leveraging the training dataset. In the former case, we sample $\mathbf{z}^{\text{bg}}$ and all $\mathbf{z}_k^{\text{app}}$ from the standard normal prior of the $\beta$-VAE, but reject objects for which the binary entropy of the mask (averaged across all

pixels) exceeds a threshold (for figures in the main paper, 100 bits). We found that empirically, this entropy threshold can be used to trade diversity of samples for higher-quality samples (cf. supplement). For the coordinates, a uniform prior within the image yields reasonable samples, and scales can be sampled from a uniform distribution between $64 \times 32$px and $192 \times 96$px and at fixed $2:1$ ratio. Alternatively, all distributions can be fit based on values obtained from the motion segmentation (object and background latents, distribution of sizes, distribution of coordinates). We provide a detailed analysis in the supplement.

## 5 EXPERIMENTS

**Motion Segmentation.** We quantify the quality of the motion segmentation approach to understand which and how many errors are propagated to later stages. Motion segmentation was scored as follows: For every frame, we matched the predicted and ground truth masks using the Hungarian algorithm (Kuhn, 1955) with the pairwise IoU (Intersection over Union) as matching cost. The frame-wise IoU scores were then aggregated as in the DAVIS-2017 benchmark (Pont-Tuset et al., 2017) by first averaging IoUs over frames for each ground truth object, and then averaging over objects. In Tab. 1 we report the segmentation performance separately for the background and the foreground objects.

The best performance is reached when using optical flow predicted by the FlowNet 2 model, thus we're using this model for all subsequent experiments. Our setting seems to be especially difficult for the ARFlow model. Different from the other networks, this model does not easily transfer from unrelated training settings and doesn't improve much when training on the Fishbowl data directly using the default hyperparameters. It might however be possible to improve the model by adapting the training setup to our dataset more closely.

Overall, the evaluation reveals two types of errors: (1) even the best model variant only detected 67.4% of all objects, i.e., one third of the objects are missed and get included in the background masks. (2) The recall decreases when increasing the IoU threshold to 0.5, i.e., the predicted masks are often not precise. While the motion segmentation therefore induces a sort of label noise, it has to be considered that not all errors are problematic for the motion segmentation. In particular, this holds for all objects that have not been detected at all and unsystematic errors.

**Object model.** To evaluate the capability of the object model to reconstruct unoccluded objects from occluded inputs, we extract object crops from the validation set using the bounding boxes of the ground truth unoccluded masks. For those crops, we compare the masks and appearances reconstructed by the model to the unoccluded ground truth (Fig. 2) using IoU and mean average error (MAE), respectively. As we are only interested in the reconstruction error within the object masks, we evaluate the MAE only on the intersection of the predicted and ground truth mask. We consider the ground truth unoccluded segmentation masks as our baseline: they correspond to a model that perfectly segments all visible object parts, but does not complete partial objects. We train variants of the object model using all augmentation strategies described above. For comparison, we also train each model variant using the ground truth occluded segmentation masks so that errors propagated from the motion segmentation can be differentiated from errors inherent to the object model.

As the results in Tab. 2 show, using another object from the input batch as artificial occlusions results in the best IoU of 0.827. When considering only those objects that are occluded at least by 50%, the IoU drops to 0.705. This performance is substantially greater than the baseline (0.27), which indicates that the model learns to complete partial objects even for those hard cases. In terms of the appearance error, using the cutout augmentation performs best. Interestingly though, for both the mask and appearance error, training without any augmentation at all reduces the performance only slightly.

When training on the ground truth segmentation instead of the motion segmentation, the error in terms of the IoU reduces by roughly one third. Performance gains can therefore be expected from improving the generative object model and, to a lesser extent, the motion segmentation. Overall, the object model seems to be quite robust to the label noise induced by the motion segmentation.

We visualize reconstructions and samples from the object model in Fig. 4. In agreement with the quantitative results before, there is a visible quality difference between the model trained on motion segmentation and ground truth segmentation, respectively. In some cases, parts of the fish such as the tail fin is missing and in other cases, the mask covers areas not properly covered by the appearance. In particular this holds for the hard cases with substantial occlusions. Moreover, the samples from the model trained on motion segmentation masks seems to contain fewer sharks (a rare class). The

Table 2: Comparison of the reconstructions from the object model with the ground truth unoccluded objects. The reconstructed masks are evaluated using the intersection over union (IoU), the appearance is evaluated using the mean average error (MAE) of the RGB values (0–255) on the intersection of the ground truth and predicted masks. Additionally, we report those metrics for hard samples in which at most half of the object is visible (IoU@0.5 and MAE@0.5). We consider model variants that use different artificial occlusions as augmentations during training. Furthermore, we compare the performance of each model variant when trained on the motion segmentation and the ground truth masks, respectively. All results are averaged over independent runs with 3 different seeds and reported with the standard error of the mean. As baseline we report the IoU of the ground truth occluded masks with the ground truth unoccluded masks.

| Training data | Augmentation | IoU ↑ | MAE ↓ | IoU@0.5 ↑ | MAE@0.5 ↓ |
|---|---|---|---|---|---|
| Motion Segmentation | None | 0.820±0.002 | 13.0±0.047 | 0.669±0.002 | 24.3±0.037 |
| | Cutout | 0.822±0.001 | **13.0±0.032** | 0.677±0.002 | 24.0±0.017 |
| | Other object | **0.827±0.001** | 14.8±0.095 | **0.705±0.001** | **23.4±0.049** |
| Ground Truth Segmentation | None | 0.885±0.001 | 12.4±0.144 | 0.738±0.002 | 17.5±0.210 |
| | Cutout | **0.887±0.001** | **12.3±0.035** | 0.743±0.003 | **17.3±0.087** |
| | Other object | 0.883±0.001 | 14.1±0.250 | **0.782±0.002** | 17.7±0.197 |
| Baseline | | 0.915 | | 0.271 | |

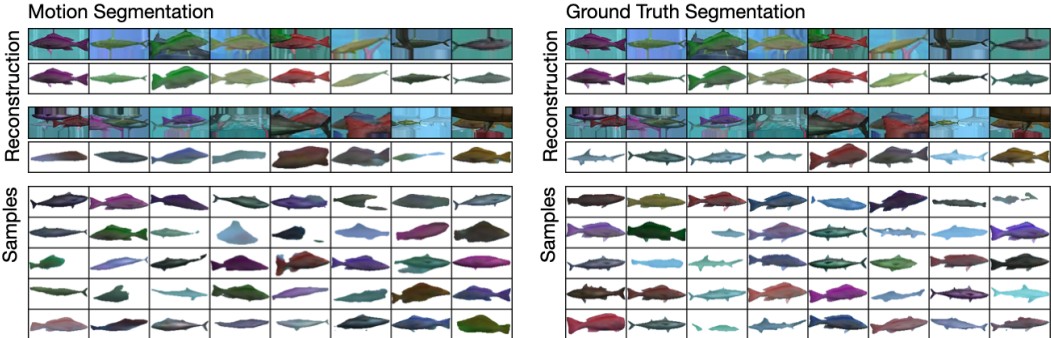

Figure 4: Qualitative results from the object model using another object as augmentation. *Left:* Object model trained on the masks predicted by motion segmentation. *Right:* Object model trained using ground truth segmentations. *Top:* Reconstructions of validation set elements. *Middle:* Reconstructions of validation set elements that are occluded by $0.5 \pm 0.05$. *Bottom:* objects sampled from the respective model. The shown reconstructions and samples are not cherrypicked and do not use entropy filtering as in the scene model.

majority of the reconstructions and samples however looks convincing, with the model capturing the relevant properties of the object masks and appearances.

**Background model.** We train two variants of the background model using the foreground/background segmentation and the ground truth segmentation masks, respectively. We show samples from the former variants as part of the scene model in Fig. 5, and provide additional samples in the supplement. We deliberately chose to not exhaustively tune the background model and defer improvements (which are conceivable given the recent advances in high resolution image modeling with VAEs) to future work. Within the scope of this work, we limit ourselves to the constrained resolution resulting in blurry samples. They still reflect the main variations in the dataset, like the overall illumination and background color.

**Scene model.** In Fig. 5 we visualize samples from the scene model following the scene statistics from the training dataset. As before, we consider two variants of the scene model using the background and object models trained on the motion segmentation and ground truth segmentation, respectively. The samples from both models nicely resemble the scenes from the training dataset. However, in both cases the errors from the object and the background model, as discussed before, are also observable in the generated scenes.

A particular strength of our modular approach is that it allows to separately intervene on causally independent mechanisms (Pearl, 2009; Schölkopf et al., 2021) of the scene generation. In Fig. 6 we show such interventions on a sample from the unsupervised scene model. The number of objects, their positions and sizes are all represented explicitly by the scene model so that we can manipulate those scene properties separately. Moreover, the background and the objects are represented

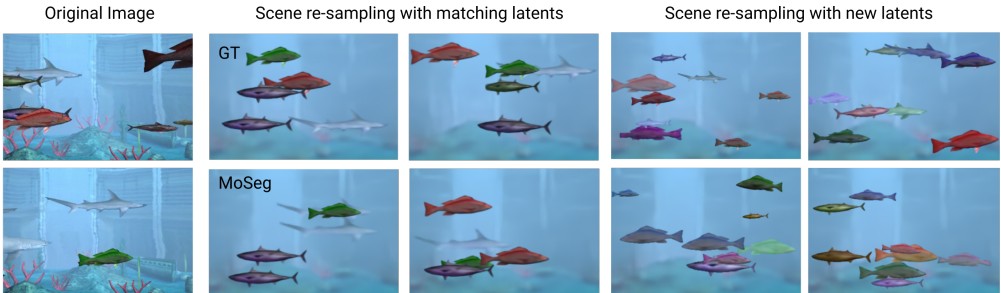

Figure 5: Samples from the training dataset in comparison to the scene model using the background and object models trained on the motion segmentation and ground truth segmentation, respectively.

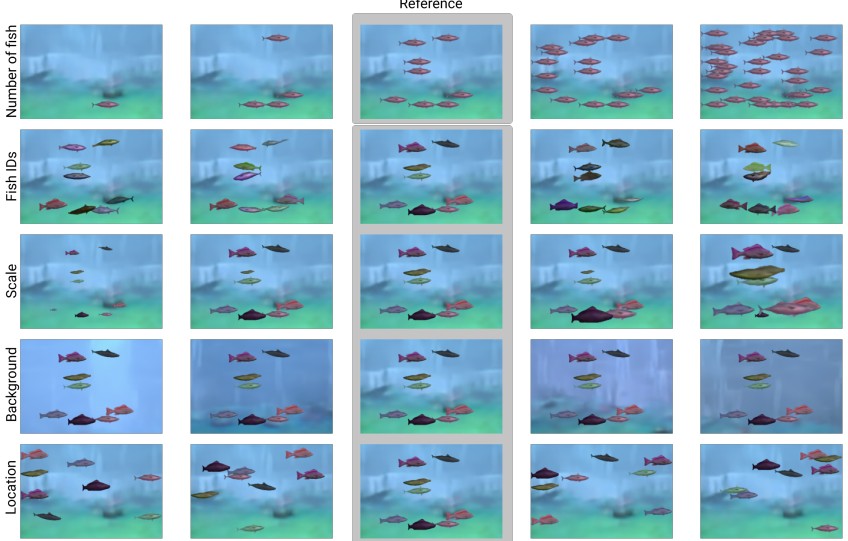

Figure 6: Interventions on the scene properties of a sample from the unsupervised scene model. Top to bottom: (i) Varying the number of objects, (ii) varying the object identities, (iii) varying the scales of individual objects, (iv) changing backgrounds while keeping objects constant, (v) re-sampling locations of all objects.

separately by their respective latent codes so that we can exchange and manipulate each of those scene components individually.

**Modularity.** A downside of end-to-end models are non-trivial dependencies between individual components, making it harder to train and debug such models. In contrast, our modular approach allows to exchange individual components in the object learning pipeline. In Figure 7, we show a proof-of-concept for using a GAN (Mescheder et al., 2018) as part of a hybrid object model (image modeling: GAN, mask modeling: VAE) within our framework.

**Comparison to prior work.** In the appendix, we further compare our approach with two recent end-to-end object-centric models (Lin et al., 2020b; Engelcke et al., 2021). Both models struggle to scale to the FISHBOWL dataset, highlighting the advantage of our multi-stage approach compared to end-to-end scene models. Moreover, we apply our method to the *RealTraffic* dataset of real world videos captured by a traffic camera, showing that our approach transfers well to other datasets.

## 6 DISCUSSION

The fields of machine learning and causal modeling have recently begun to cross-pollinate. Causality offers a principled framework to think about classes of distributions related to each other by domain shifts and interventions, and independent/modular components (e.g., objects) that are invariant across such changes. Vice versa, machine learning has something to offer to causality when it comes to learning causal variables and representations—traditionally, research in causal discovery and inference built on the assumption that those variables be given a priori. Much like the shift from classic AI towards machine learning included a shift from processing human-defined symbols towards automatically learning features and *statistical* representations, causal representation learning aims to learn *interventional* representations from raw data (Schölkopf et al., 2021). It is in this

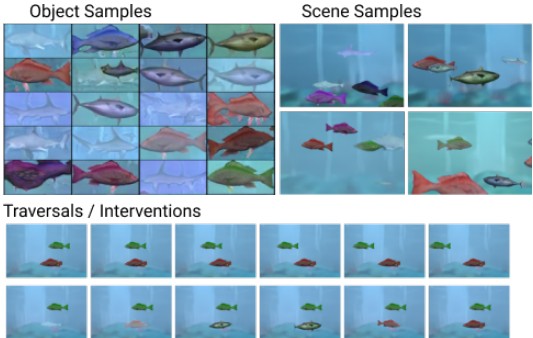

Figure 7: The modularity of our multi-stage approach allows to easily exchange individual components. Here, we replace the image-modeling part of the object model with a GAN instead of the previously used VAE. As in our main model, this allows meaningful traversals/interventions since objects and their positions/sizes are presented as individual entities.

sense that we like to think of our scene model as a causal generative model, even though it is fairly straightforward from the causal point of view.

We propose a multi-stage approach for learning generative object models from unlabelled videos. Inspired by the Gestalt *Principle of Common Fate*, we use candidate segmentations obtained from unsupervised optic flow to weakly supervise generative object and background models, and combine them into a scene model. The scene model is causal/mechanistic in the sense that it (1) learns complete objects (although occlusions are abundant in the training data)[2] and (2) properly handles partial depth orderings in the scene, the latter through the dead leaves approach—as opposed to the computationally simpler additive mixture approach which is common place in end-to-end scene models, but contradicts the physics of scene generation. Moreover, while many existing models only decompose and represent scenes, (3) our model can generate plausible novel and out-of-distribution scenes (e.g., higher fish density, different sizes). This showcases the challenges faced by end-to-end learning when not endowed with sufficient prior structure and inductive biases.

While learning object models as part of an end-to-end trainable object centric scene model might be more elegant in principle, it has not been shown to scale to more realistic data yet. We instead chose a multi-stage approach that allows developing and analyzing the modeling stages individually. This approach allowed us to conclude that exploiting motion information for object discovery, as done in the motion segmentation stage, is a valuable cue that greatly simplifies object learning. Also, since the scene model uses the component models as modelling "atoms", this naturally permits intervening on the parameters defining the scene composition without side-effects, as stipulated by the principle of Independent Causal Mechanisms (Parascandolo et al., 2018; Peters et al., 2017).

We evaluated the modelling stages on the novel FISHBOWL dataset, consisting of short videos recorded from a synthetic 3D aquarium. Notably, we found that the generative object model proves to be fairly robust to errors in the motion segmentation. We expect further performance gains to be achievable by improving the visual fidelity of the generative object and background models (Razavi et al., 2019; Vahdat & Kautz, 2020); however, this was not the motivation for the present work.

We see several ways to extend our model in the future, beyond improving the implementations of the individual stages by taking advantage of progress in computer vision. One limitation of using motion segmentation is that we can only decompose scenes of moving objects. This could be addressed by integrating object models trained using our approach into an end-to-end scene model, e.g., the work of Eslami et al. (2016) and its successors. When using a learning based motion segmentation approach, there could be a feedback circle with both models improving each other as training progresses and this way closing the observed gap to using ground truth segmentations. Finally, our object model does currently not capture object motions. Given that the motion segmentation yields temporally consistent segmentation masks, this could, in principle, well be included in our approach.

---

[2]since inpainting takes place at the level of masks/shapes, rather than only for pixels as in most prior work.

**Ethics statement.** While there is some way to go when it comes to the generation of photorealistic scenes, we note that being able to automatically generate coherent scenes, including the interventional control that our approach pioneers, could be misused to synthesize or modify scenes, reinforcing the need to develop methods that allow to reliably detect manipulated content. On the other hand, the interpretability that comes with the modularity of our method can yield insights into scene perception, and could therefore facilitate better *explanations* for model decisions.

**Reproducibility statement.** We took several steps to make our work reproducible: For the dataset generation, we modified the Aquarium simulation on which our dataset is based, in order to make the rendering fully deterministic. The configuration files describing the recording settings for all samples will be published with the generation code along the dataset, which allows re-recording all videos. Moreover, the code for training our models is deterministic and will be made publicly available.

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

## A    ADDITIONAL DETAILS ABOUT THE METHOD

### A.1    TRAINING ARFLOW ON FISHBOWL

We build on the official implementation of ARFlow provided in https://github.com/lliuz/ARFlow. To train the model on the Fishbowl dataset, we make the following adaptations to the original configuration used for training on Sintel:

- The internal resolution of the model is set to 320x448 pixels.
- For training, we use the first 200 videos of the Fishbowl dataset. This amounts to 25.4K frame pairs which is substantially more than the Sintel dataset (pretraining: 14,570, main training: 1041), which is the largest dataset on which ARFlow was originally trained. Initial experiments using the first 1000 videos did not lead to an improvement in the training objective compared to only using 200 videos.
- We train the model using a batch size of 24 and use 300 random batches per epoch. We perform both the pretraining and main training stage, but using the same data for both stages. The pretraining stage is shortened to 100 epochs, after which the training loss did not improve further.

We selected above parameters by pilot experiments using the final training loss as criterion. All other hyperparameters, in particular regarding the losses, the augmentations and the model architecture, are used unchanged. We remark that the hyperparameters are chosen differently for the two datasets used in the original paper, so we conjecture that the performance on this model most likely improves when closely adapting the training scheme to our setting.

### A.2    OBJECT MODEL

We loosely build on the $\beta$-VAE implementation by Subramanian (2020). We reimplemented the training script and modified architecture details like the latent dimensionality due to differences in the image size.

We use a CNN with 10 layers as an encoder. Each layer consists of a $3 \times 3$ convolution, followed by layer normalization (Ba et al., 2016) and a leaky ReLU nonlinarity with a negative slope of 0.01. The decoders are built symmetrically by using the reversed list of layer parameters and transposed convolutions. The decoders both use an additional convolutional decoding layer, without normalization and nonlinearity. The detailed specification of the default hyperparameters used by the object model is given in the following table:

Table 3: Default hyperparameters used for the object model.

| Parameter | Value |
|---|---|
| sample size | $128 \times 64$ |
| hidden layers: channels | 32, 32, 64, 64, 128, 128, 256, 256, 512, 512 |
| hidden layers: strides | 2, 1, 2, 1, 2, 1, 2, 1, 2, 1 |
| latent dimensions | 128 |
| prior loss weight ($\beta$) | 0.0001 |
| mask loss weight ($\gamma$) | 0.1 |
| learning rate, epoch 1-40 | 0.0001 |
| learning rate, epoch 41-60 | 0.00001 |

We chose the parameters regarding the architecture based on early experiments by qualitatively evaluating the sample quality. The learning rate and mask loss weight $\gamma$ where determined using grid searches with the IoU of the mask as selection criterion. However, we noticed a high degree of consistency between the rankings in terms of the mask IoU and appearance MAE. The best reconstruction quality was obtained when not regularizing the prior (i.e., $\beta = 0$). We chose the final value of $\beta = 0.0001$ as a compromise between reconsutruction and sampling capabilites based on visual inspection of the results.

For the mask and appearance losses defined in the main paper, we use an implementation based on the following PyTorch-like pseudo code:

```
def object_model_loss(image, mask_fg, mask_bg, gamma = 1., beta = 0.001):
  # image (b,c,h,w), mask_fg (b,h,w), mask_bfg (b,h,w)
```

```
3
4    latents = encode(image)
5    mask_pred, img_pred  = mask_decoder(latents), img_decoder(latents)
6
7    L_img  = (mask_fg * mse(img_pred, image)).sum()
8    L_mask = ((mask_fg + mask_bg) * bce(mask_pred, mask_fg)).sum()
9    L_reg  = kl_divergence(latents)
10   Z_img, Z_mask = mask_fg.sum(), (mask_fg + mask_bg).sum()
11
12   return L_img / Z_img + gamma * L_mask / Z_mask + beta * L_reg
```

### A.3 BACKGROUND MODEL

**Training details**  We use a similar $\beta$-VAE and training objective for the background model as for the object model. Different to the object model, we do not predict the mask and instead only reconstruct a non-occluded background sample. We sweep over learning rates in $\{1 \cdot 10^{-3}, 5 \cdot 10^{-4}, 1 \cdot 10^{-4}\}$ and $\beta$ in $\{10^{-2}, 10^{-3}, 10^{-4}, 10^{-3}\}$ and select the model with $\beta = 10^{-3}$ and learning rate $10^{-4}$ which obtained lowest reconstruction and prior loss. A higher value for $\beta$ caused the training to collapse (low prior loss, but high reconstruction loss). As noted in the main text, the background model performance (and resolution) can be very likely improved by using larger models, more hyperparameter tuning and better decoders suited to the high resolution. As noted in the main paper, we omit these optimizations within the scope of this work and rather focus on the object and scene models.

**Implementation**  The background model loss is similar to the reconstruction loss of the foreground model. We omit reconstructing the mask. In the input image for the background model, all foreground pixels are replaced by the average background RGB value to avoid distorting the input color distribution, refer to the example images in Fig. 11. The loss can be implemented as follows:

```
1  def background_model_loss(image, mask_bg, beta = 0.001):
2    # image (b,c,h,w), mask_fg (b,h,w), mask_bfg (b,h,w)
3
4    latents  = encode(image)
5    img_pred = img_decoder(latents)
6
7    L_img = (mask_bg * mse(img_pred, image)).sum()
8    L_reg = kl_divergence(latents)
9    Z_img = mask_bg.sum()
10
11   return L_img / Z_img + beta * L_reg
```

### A.4 SCENE MODEL

**Mask temperature**  By default, the mask is computed by passing the logits obtained from the object mask decoder through a sigmoid non-linearity for obtaining probabilities. During scene generation, we added sharpness to the samples by varying the temperature $\tau$, yielding an object mask

$$\mathbf{m} = \frac{1}{1 + \exp(-\mathbf{x}/\tau)}, \tag{1}$$

where $\mathbf{x}$ is the logit output by the mask decoder and the exponential is applied element-wise. "Cooling" the model to values around $\tau = 0.1$ yields sharper masks and better perceived sample quality. Note that the entropy threshold introduced in the next section depends on the value of $\tau$.

**Entropy filtering**  When sampling objects during scene generation, we filter the samples according to the model "confidence". A simple form of rejection sampling is used by computing the mask for a given object latent, and then computing the entropy of that mask. Given a mask $\mathbf{m}(\tau)$, we adapt the sampling process considering the average entropy across all pixels in the mask,

$$H_2(\mathbf{m}) = -\frac{1}{WH} \sum_{u=1}^{W} \sum_{v=1}^{H} \mathbf{m}(u,v) \log_2 \mathbf{m}(u,v) + (1 - \mathbf{m}(u,v)) \log_2 (1 - \mathbf{m}(u,v)) \tag{2}$$

and reject samples where $H_2(\mathbf{m})$ exceeds a threshold. Reasonable values for a mask temperature of $\tau = 0.1$ are around 100 to 200 bits. It is possible to trade sample quality and sharpness for an increased variability in the samples by increasing the threshold.

## B  ADDITIONAL DETAILS ABOUT THE DATASET GENERATION

A demo of the original aquarium WebGL demo used for our experiments is available at webglsamples.org/aquarium/aquarium.html. The original source code is released under a BSD 3-clause license (Google Inc.) and available at github.com/WebGLSamples/WebGLSamples.github.io.

We apply the following modifications to the original demo for our recording setup:

- We implemented a client-server recording setup: The client pulls recording parameters from the server and sends back the recorded frames with ground truth segmentation to the server. The server maintains the overall list of randomly sampled aquarium configurations and post-processes the recordings into their final form. We run this setup in parallel on multiple nodes, using the headless Chrome browser[3].
- All samples are recorded for 128 frames at 30Hz. We modified the aquarium to only advance to the next frame once the current frame is captured and sent to the server.
- For recording the masks, we modified the texture shaders to use a unique color for rendering every object. Furthermore, we disabled all rendering steps modifying those colors (anti-aliasing, alpha compositing).
- To make the recording fully deterministic, we disabled the bubbles in the aquarium.
- To increase the diversity of the fish in the aquarium, we applied random color shifts to all fish textures except sharks, for which those color shifts look very unnatural. We shifted each color by rotating each color value in the IQ plane of the YIQ color space. For each fish, we independently and uniformly sampled one of 8 discrete color shifts regularly spaced in $[0°, 360°]$.
- During post-processing, we used ffmpeg[4] to rescale and centrally crop $480 \times 320$px from every frame and encode all frames as a video. We cropped and rescaled the masks same way, but exported them as json using the run length encoding provided by the COCO API[5].
- For recording the unoccluded ground truth, we used above setup to record and post-process frames and masks of the validation and test samples, but only render the respective fish of interest.

The code used to generate the dataset including all parameters will be made publicly available.

---

[3]https://github.com/puppeteer/puppeteer/
[4]https://ffmpeg.org
[5]https://github.com/cocodataset/cocoapi

## C ADDITIONAL RESULTS

### C.1 OBJECTS EXTRACTED BY THE MOTION SEGMENTATION

The following figure shows examples of objects extracted from the original videos using motion segmentation and the ground truth occluded masks, respectively. The figure reveals typical failure modes, such as multiple fish that are segmented jointly, and parts of the background contained in the object mask. Most of the fish however, are segmented very accurately.

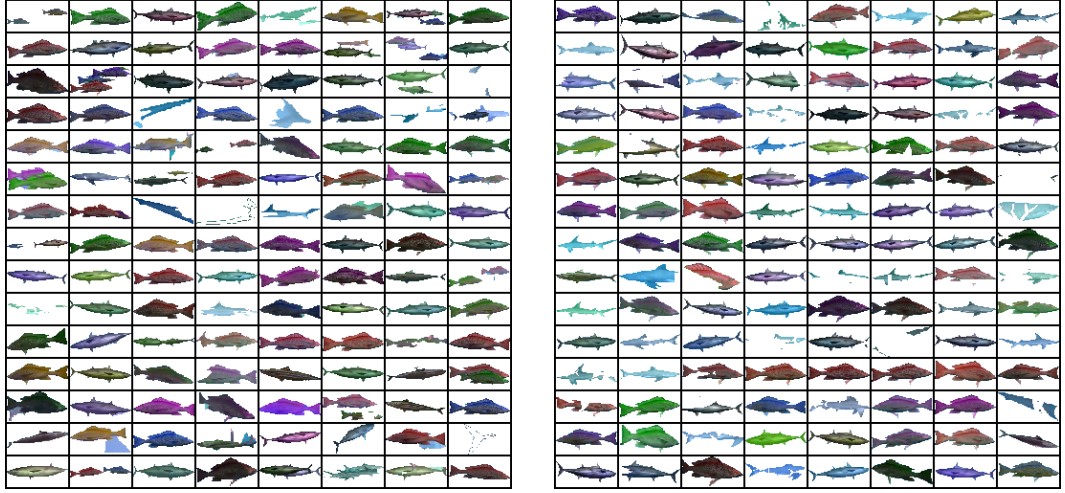

Figure 8: Objects extracted from the training videos that are used for training the object model. *Left:* Objects extracted using the motion segmentation. *Right:* Objects extracted using the ground truth occluded segmentation masks.

## C.2 Object model: additional reconstructions

In Fig. 9 we show additional reconstruction from the object model trained on motion segmentation and ground truth occluded masks, respectively. Moderate occlusion levels up to 30% are handled well by both variants. At higher noise levels however, only the variant of the object model trained on the ground truth masks is able to correctly complete the partial objects.

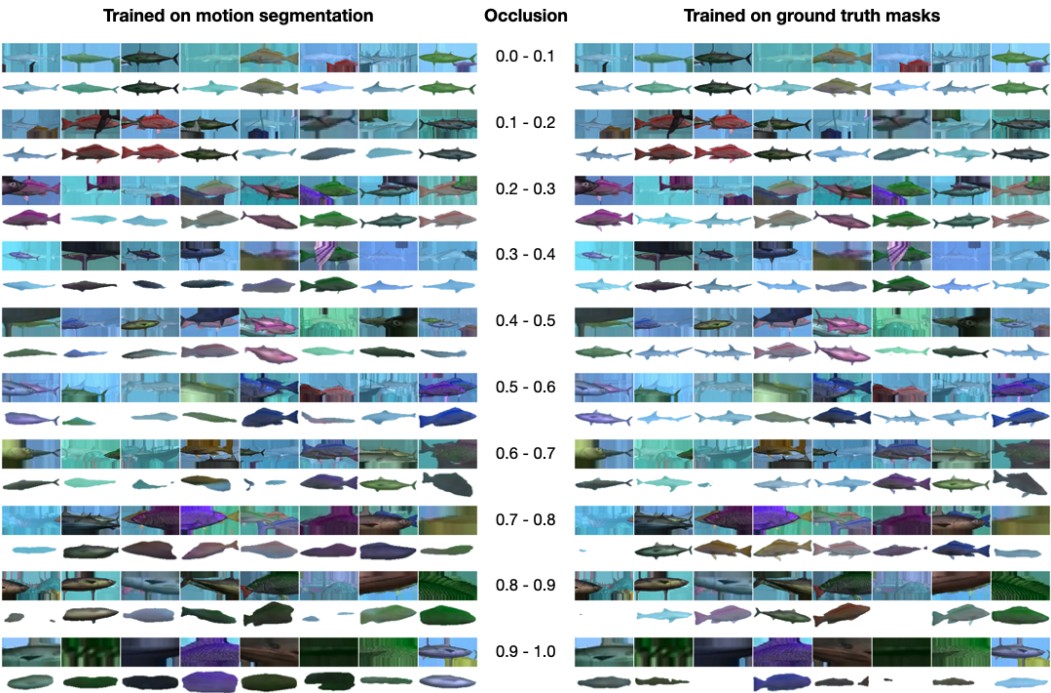

Figure 9: Reconstructions from the object model for input crops with different occlusion levels (0.0 = no occlusion, 1.0 = fully occluded). For each occlusion level, the input images and the respective reconstructions are shown. Both model variants are trained using another fish as artificial occlusion during training. *Left:* Object model trained using the motion segmentation masks. *Right:* Object model trained using the ground truth unoccluded masks.

## C.3 OBJECT MODEL: ADDITIONAL SAMPLES

The following figure shows additional samples from the object models trained using another object as artificial occlusion (the same models as used for Fig. 4 in the main paper).

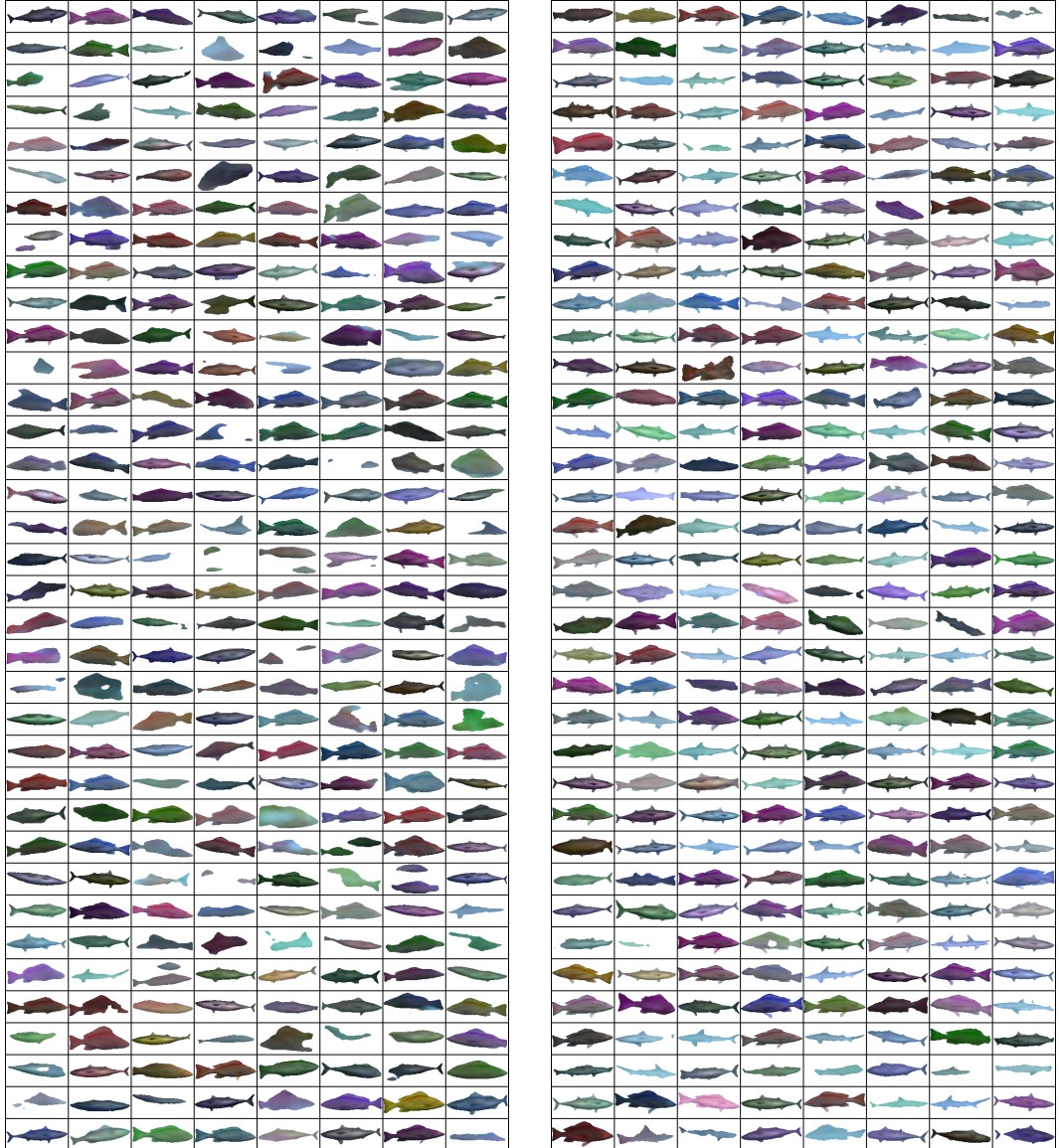

Figure 10: Samples from the object model using another input object as augmentation during training. These are the same models as used for Fig. 4 in the main paper. *Left:* Object model trained on the motion segmentation. *Right:* Object model trained on the ground truth occluded masks.

## C.4 BACKGROUND MODEL: INPUTS

The background model is trained on images pre-processed by an ensembling of foreground-background algorithms from Sobral (2013). We use $\Sigma - \Delta$ (Manzanera & Richefeu, 2007), static frame differences, LOBSTER (St-Charles & Bilodeau, 2014) and PAWKS (St-Charles et al., 2016) with standard hyperparameters set in Sobral (2013). The goal behind this choice is to detect as many objects as possible and remove the amount of erroneously included foreground pixels — it might be possible to even further improve this pre-processing step with different algorithms and hyperparameter tuning.

We give a qualitative impression of the background input samples in Fig. 11. Note that all foreground pixels were replaced by the average color value obtained by averaging over background pixels to avoid inputting unrealistic colors into the $\beta$-VAE.

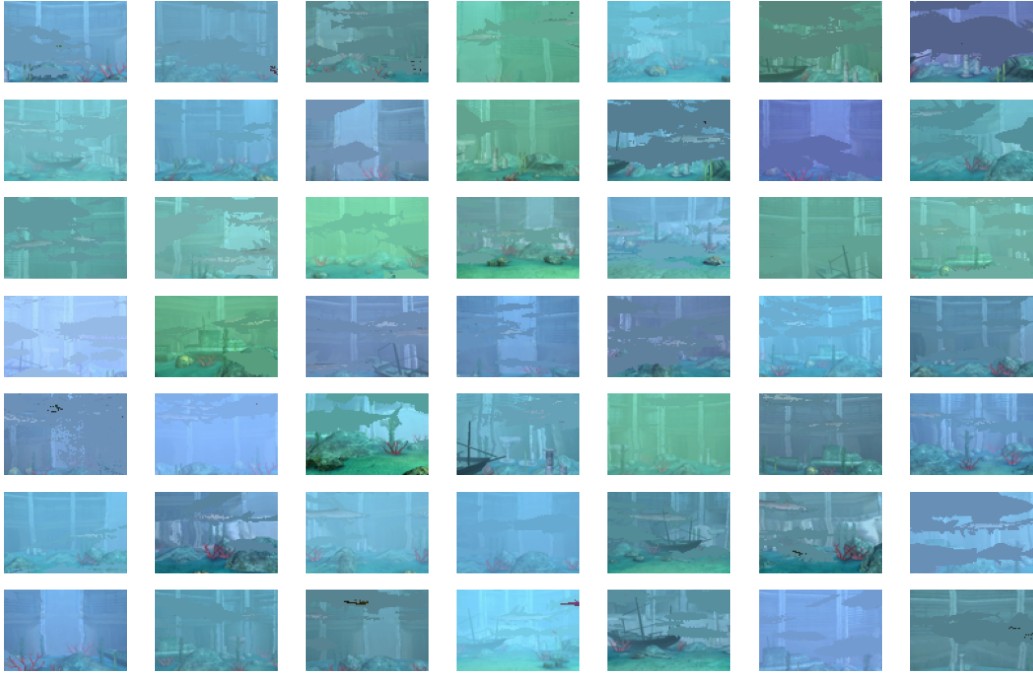

Figure 11: Input samples to the background $\beta$-VAE. Objects were removed by applying an ensemble of foreground-background segmentation algorithms. Images are resized to $96 \times 64$px to trade-off training speed for sample quality.

## C.5 ADDITIONAL SAMPLES FROM THE SCENE MODEL

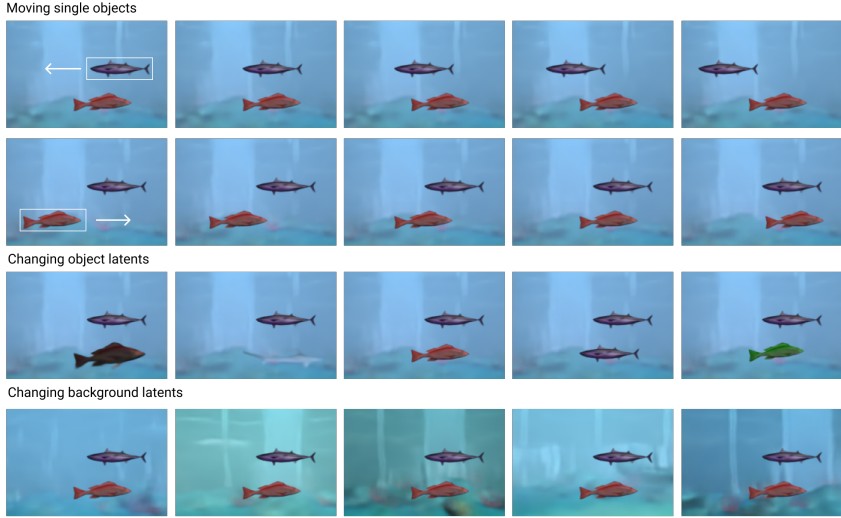

Figure 12: Two object "reconstructions". Object latents are obtained from a reference sample. Our modular scene model makes it straightforward to vary locations of single objects, exchanging single objects, or changing the background without affecting the output sample (top to bottom).

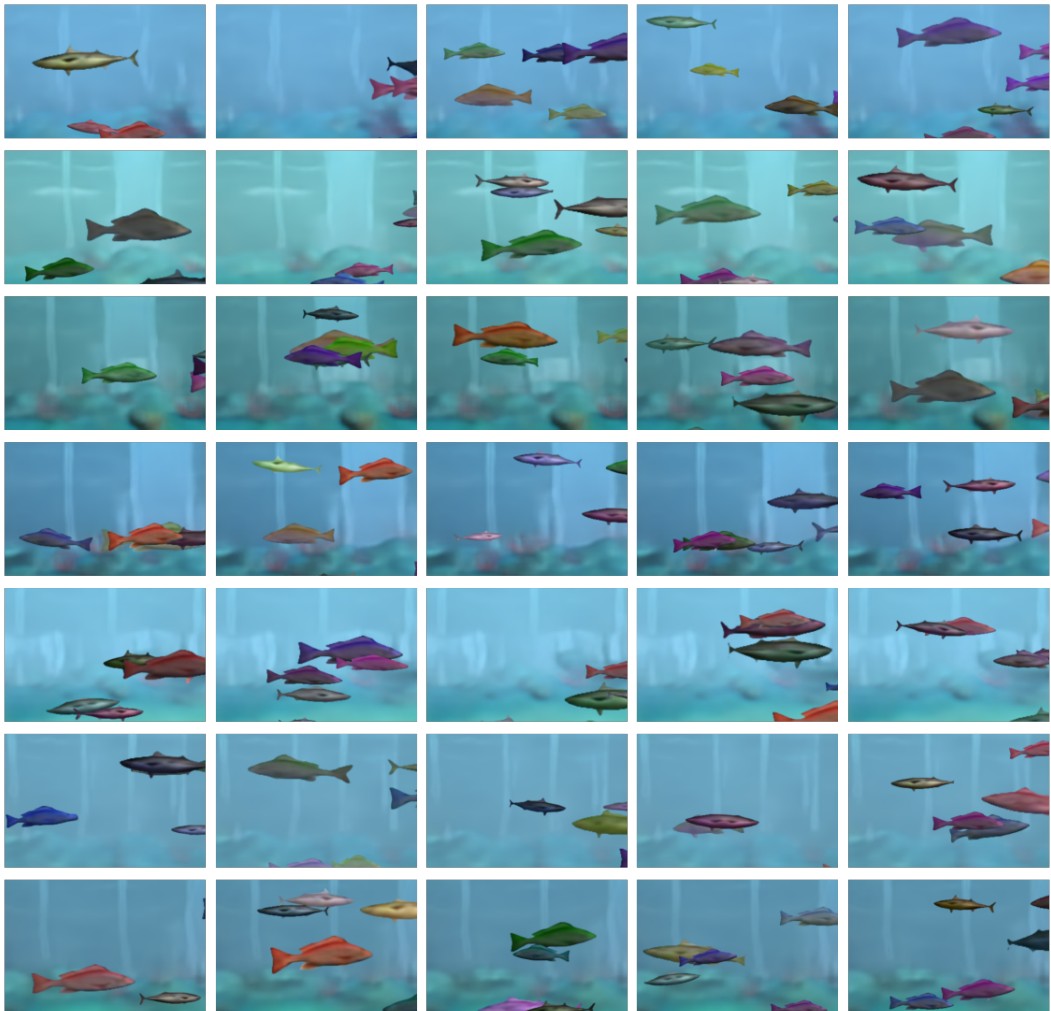

Figure 13: Additional samples from the scene model. Depicted samples use different (reconstructed) backgrounds, samples from the object model are obtained from the standard normal prior and filtered with 150 bit entropy threshold at $\tau = 0.2$, sample sizes are constrained on a reference training sample, object positions are sampled independently from a uniform prior. Samples are not cherrypicked.

## C.6 CONDITIONAL SAMPLING FROM THE SCENE MODEL

We present conditional samples from the scene model. As a simple baseline, we use a k-nearest neighbour approach for conditionally sampling object latents based on background latents. First, we extract a paired dataset of background and foreground latents from the training dataset. Second, we sample background latents from the standard Normal distribution (prior of the background model). Third, we compute the 2% nearest neighbouring videos based on the $L^2$ distance from the background latent. Fourth, we randomly sample a subset of foreground latents extracted by the motion segmentation. Finally, we reconstruct the scene using the background latents along with the chosen subset of foreground latents. We reject foreground latents with an entropy larger than 150 bit. Samples (non-cherrypicked) are depicted in Fig. 14

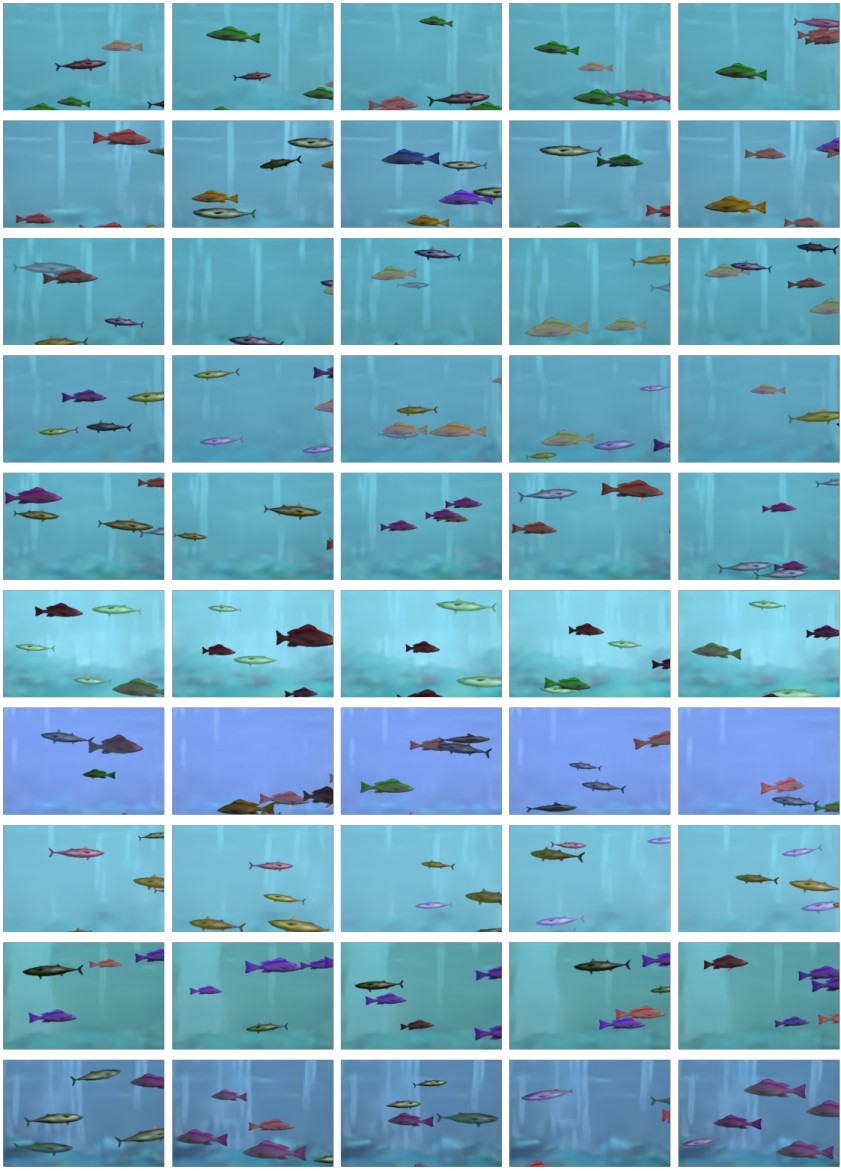

Figure 14: Conditional samples from the model. Fish appearances (qualitatively, mainly the brightness) now vary according to the background sample.

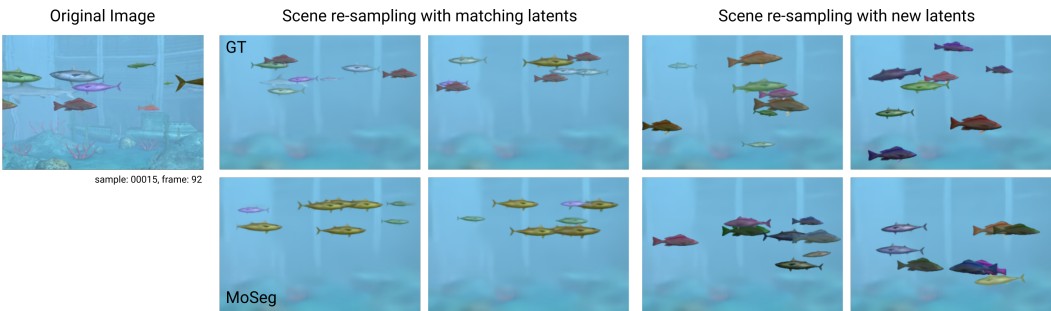

Figure 15: Conditional sampling of object locations based on a reference scene. For matching latents sampling, we extract the object latents and positions from the reference scene using the motion segmentation model. Samples obtained by matched sampling are more similar to the reference scene than samples obtained by unconstrained sampling from the model priors.

Similarly, we can constrain other latents like the x and y positions to a particular background. In Fig. 15 we show examples of constraining object locations based on a reference sample for both the ground truth and motion segmentation object model.

## C.7 ENTROPY SAMPLING

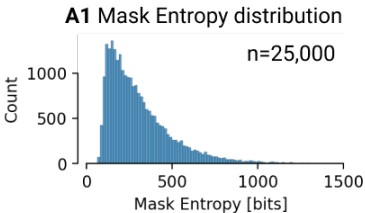

**A1** Mask Entropy distribution

n=25,000

**A2** Predicting Entropy from latents

Train R² ~0.46 (n=5,000)
Test R² ~0.45 (n=20,000)

Model: Linear Regression, 128d

**B** Top 64/1000 samples in terms of Mask Entropy

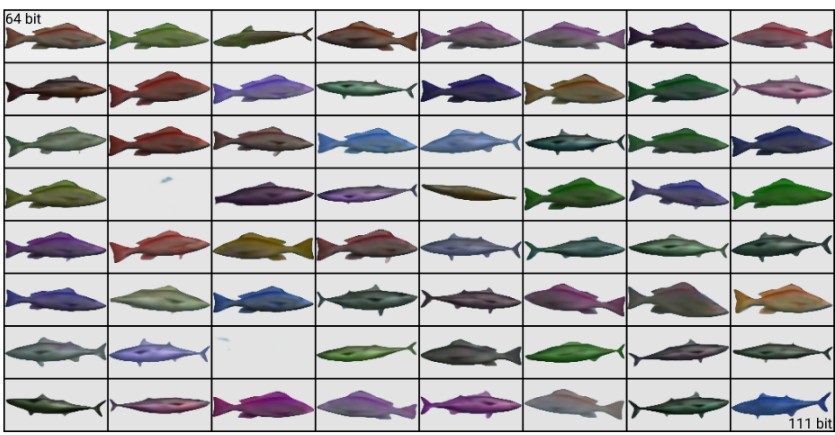

**C** Worst 64/1000 samples in terms of Mask Entropy

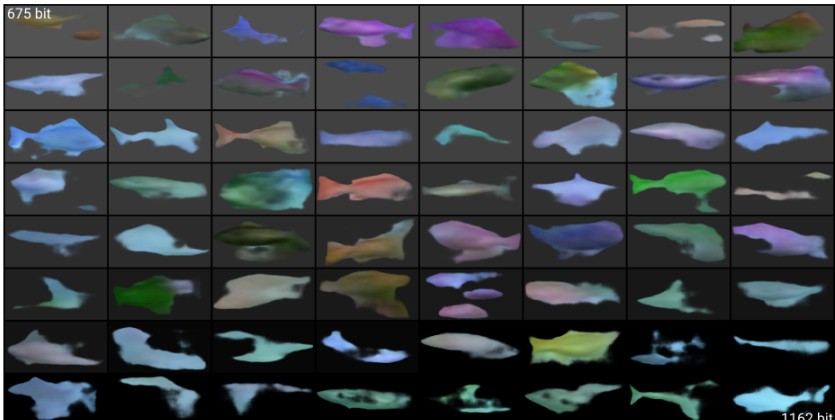

Figure 16: Details of entropy filtering when using the object model. (A1) Distribution of mask entropies for a given foreground model, estimated over 25,000 randomly sampled objects. The distribution typically peaks around 100 to 200 bits, makeing this a reasonable range for picking the cut-off. (A2) The 128d latent vector of objects is reasonably correlated with the entropy ($R^2 = 0.45$), making it possible to alter the prior to sample from to encourage low entropy samples. (B) 64 samples with the lowest entropy after drawing 1000 random samples from the object models and sorting according to entropy. While this strategy encourages some non-plausible, low entropy objects (row 4 and 6), it generally filters the dataset for samples with sharp boundaries and good visual quality. (C) 64 samples with the highest entropy after drawing 1000 random samples from the object models and sorting according to entropy. With a few exceptions of plausible samples (e.g. in row 2), most of the samples should be rejected in the scene model.

# D    COMPARISON TO RELATED METHODS

## D.1    GAN BASELINE

As a baseline method for generating novel scenes for the FISHBOWL dataset, we consider the GAN used by Mescheder et al. (2018) for the CelebA-HQ dataset (Karras et al., 2018). We used the official implementation available at https://github.com/LMescheder/GAN_stability, and only changed the resolution of the generated images to $192 \times 128$px. For training, we used every 16th frame from each input video in the training set rescaled to the resolution of the GAN, resulting in 160k training images. We trained the model for 90 epochs.

Fig. 17 shows samples from the model in comparison to training frames. Overall, the GAN is able to generate images of convincing quality resembling the training images well. A particular strength of the GAN in comparison to our method, is it's ability to generate backgrounds with many details—which we explicitly left for future work. Several fish generated by the GAN however look distorted, in agreement with previous works concluding that GANs are good at generating "stuff", but have difficulties generating objects Bau et al. (2019). This becomes especially apparent when visualizing interpolations in the latent space between samples, as done in Fig. 18.

Many differences between the samples generated by the GAN and our method, respectively, stem from the GAN being able to learn the overall statistics of the scenes well, but not learning a notion of objects. Compared to the GAN baseline, our object-centric approach offers several conceptual advantages: (1) The background and the objects in each scene are represented individually by our method, which makes it straightforward to intervene on the generated samples in a meaningful way (Fig. 6). While directions in the latent space of the GAN that correspond to semantically meaningful changes in the image might well exist, the GAN framework does not offer a principled way to find those directions without supervision. (2) While the GAN is able to generate novel scenes, it cannot be used to infer the composition of input scenes. (3) An optimally trained GAN perfectly captures the statistics of the input scene—making it impossible to generate samples beyond the training distribution. As our model explicitly represents scene parameters such as the number and position of fish, our model can be used for controlled out-of-distribution sampling (Fig. 6).

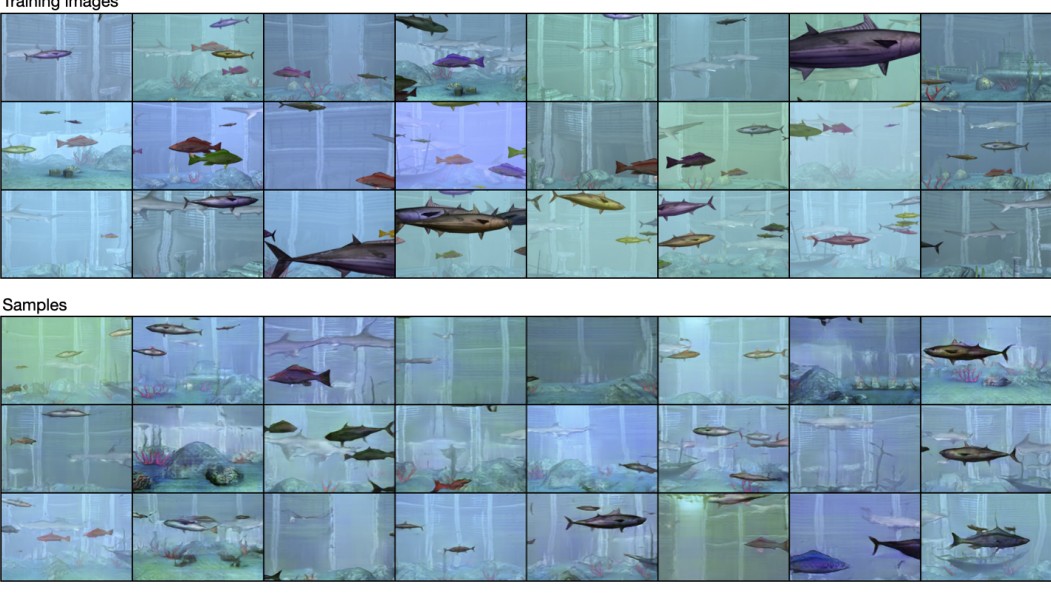

Figure 17: Samples from a baseline GAN (Mescheder et al., 2018) trained on frames from the training set of the FISHBOWL dataset. *Top:* Input frames from the dataset. *Bottom:* Samples generated by the GAN.

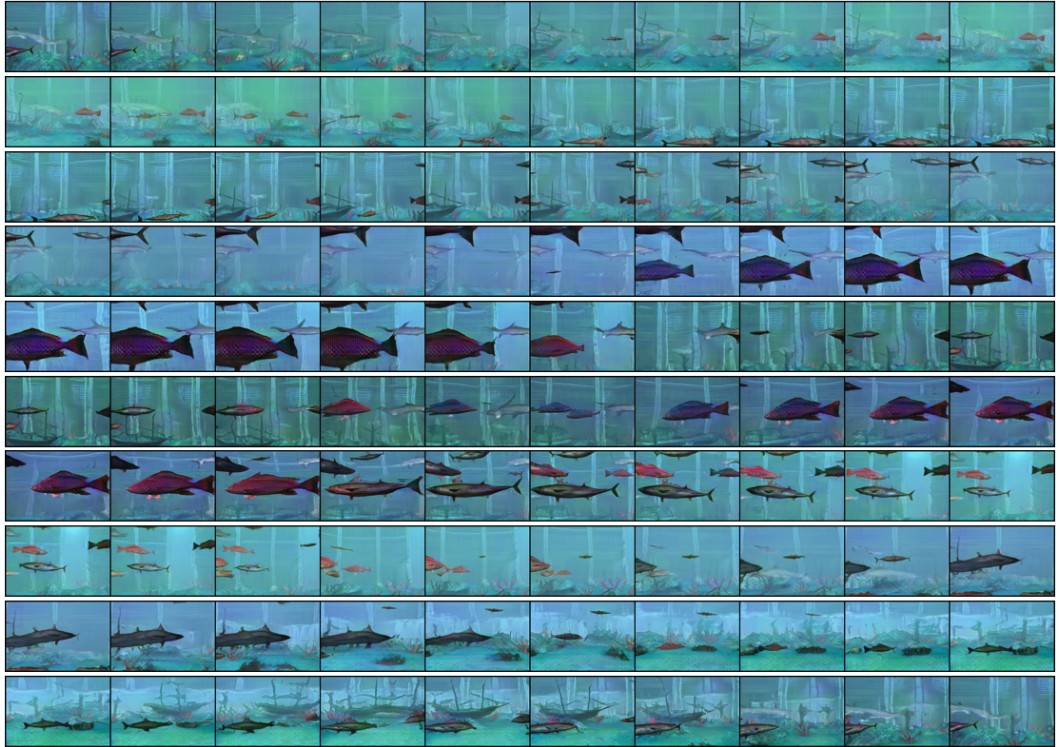

Figure 18: Samples from the baseline GAN, obtained by interpolating the latent vectors between random samples.

## D.2 SPACE (LIN ET AL., 2020B)

SPACE (Lin et al., 2020b) is an object-centric representation learning method following the Attend-Infer-Repeat (AIR) framework (Eslami et al., 2016; Crawford & Pineau, 2019). Different from the AIR model, the detection of objects and inference of their latent representation is fully parallelized to make the approach faster and more scalable. We trained SPACE on the Fishbowl dataset using the implementation provided by the authors (https://github.com/zhixuan-lin/SPACE). We used the default hyperparameters and trained two variants using a 4x4 and 8x8 grid of object detectors, respectively. As for the GAN baseline in appendix D, we used every 16th frame for training this model to keep the training time reasonable (160k frames in total). Despite the subsampling, this is still substantially more data than the model was trained on in the original paper (60k images). SPACE expects the input images to have equal width and height, therefore we used a central square crop from every frame.

Training the model on a single Nvidia RTX 2080ti GPU is relatively fast (14h for the 4x4 grid and 18h for the 8x8 grid), confirming the performance advantage of the parallel object detectors. As the results in the Fig. 19 show, the object reconstructions from the model overall look reasonable. Most structure in the background is missed by the model, however we conjecture that this might be solvable by adapting the capacity of the background network. The visualization of the object detections however reveals a more fundamental failure mode due to the grid structure used by the object detector in SPACE: Larger fish are split across several cells, even when using only 4x4 cells. As each cell can only handle at most one object, decreasing the cell count further is not expected to yield sensible results, as this limits the number of objects too much. Lin et al. (2020b) mentioned this problem and introduced a boundary loss to address it, however on the Fishbowl dataset this does not resolve the problem. We hypothesize that an approach based on a fixed grid, while adequate in some cases and having substantial performance advantages, doesn't work well with objects showing large scale variations as present in our newly proposed dataset. We believe this is a limitation of SPACE that cannot be resolved with further model tuning.

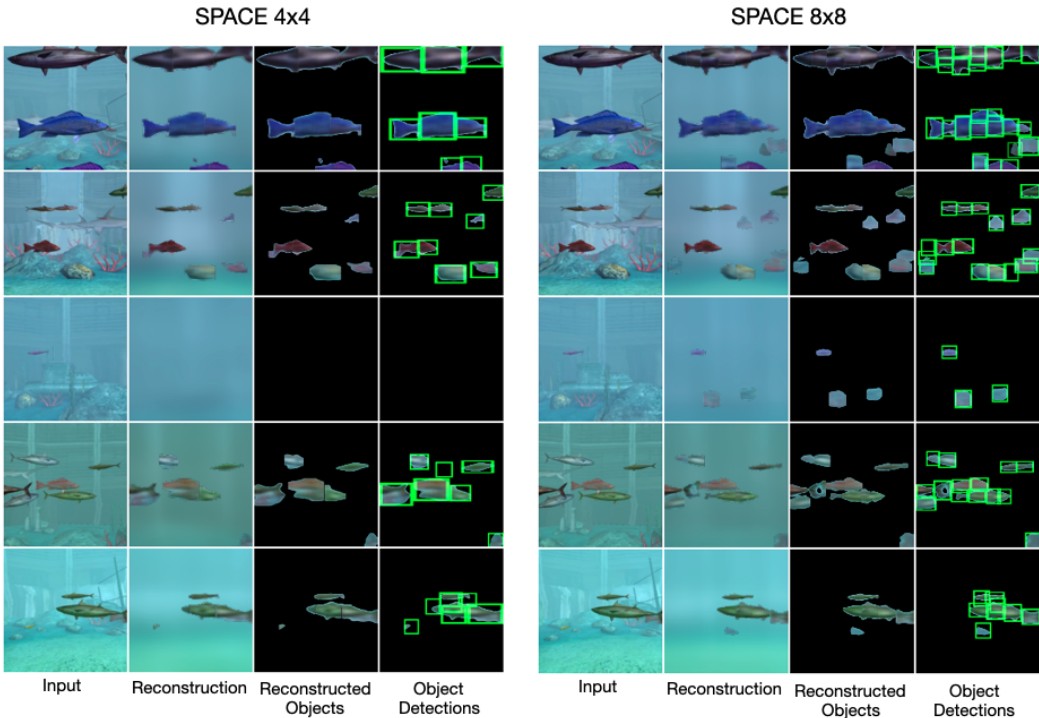

Figure 19: Scenes from the Fishbowl dataset reconstructed by SPACE (Lin et al., 2020b). The model is trained using the official implementation provided by the authors with the default parameters. The two variants shown use grid sizes of 4x4 and 8x8, respectively.

We performed an additional experiment that integrates our approach with SPACE: As a first step, we trained a variant of our object model that uses the glimpse encoder and decoder from SPACE. We increased the learning rate to 0.0005 following an initial grid search, but otherwise left all hyperparameters unchanged. As the second step, we included the pretrained glimpse encoder and decoder into the SPACE model and trained the full model end-to-end as described above (using a grid size of 4x4). When we continued to train the glimpse encoder and decoder in the end-to-end setting, we noticed similar failures as described above. Therefore we froze the respective weights after the pretraining step.

Figure 20 shows results from this model variant. Early during the training, the model is in many cases able to detect the objects in the input scenes. This especially holds for larger objects that have been shown to be problematic for the end-to-end model before. Later during training SPACE however increasingly uses the background model to also explain foreground objects. In addition, we evaluated both the pretrained and end-to-end trained object model from SPACE using the ground truth unoccluded objects. As the quantitative results in Table Tab. 4 show, the reconstruction of the appearance works better with the SPACE object model than with our architecture[6], but has a substantial gap in the segmentation performance. More importantly, the results clearly show an improvement of the motion segmentation based training over the original end-to-end approach, confirming the qualitative results.

Overall those results show that our motion segmentation based approach for object-centric modeling can also be used for scaling existing architectures to more complex settings. The off-the-shelf SPACE model however still falls short of our scene model when trained this way. In the future we see great potential in combining our object learning approach with state-of-the-art object-centric scene models.

---

[6]To some degree this is confounded with the worse segmentation performance, as the appearance error is only evaluated on the intersection of the ground truth and predicted mask.

Table 4: Quantitative evaluation of the SPACE object model trained end-to-end within SPACE and pretrained based on motion segmentations using the same loss as our object model. The pretrained SPACE object model and ours use the cutout augmentation. The metrics are the same as in Tab. 2.

| Object model | IoU ↑ | MAE ↓ | IoU@0.5 ↑ | MAE@0.5 ↓ |
|---|---|---|---|---|
| SPACE (end-to-end) | 0.533 | 12.3 | 0.425 | 16.4 |
| SPACE (pretrained) | 0.734 | **7.60** | 0.605 | **12.4** |
| Ours | **0.822** | 13.0 | **0.677** | 24.0 |
| Baseline | 0.915 | | 0.271 | |

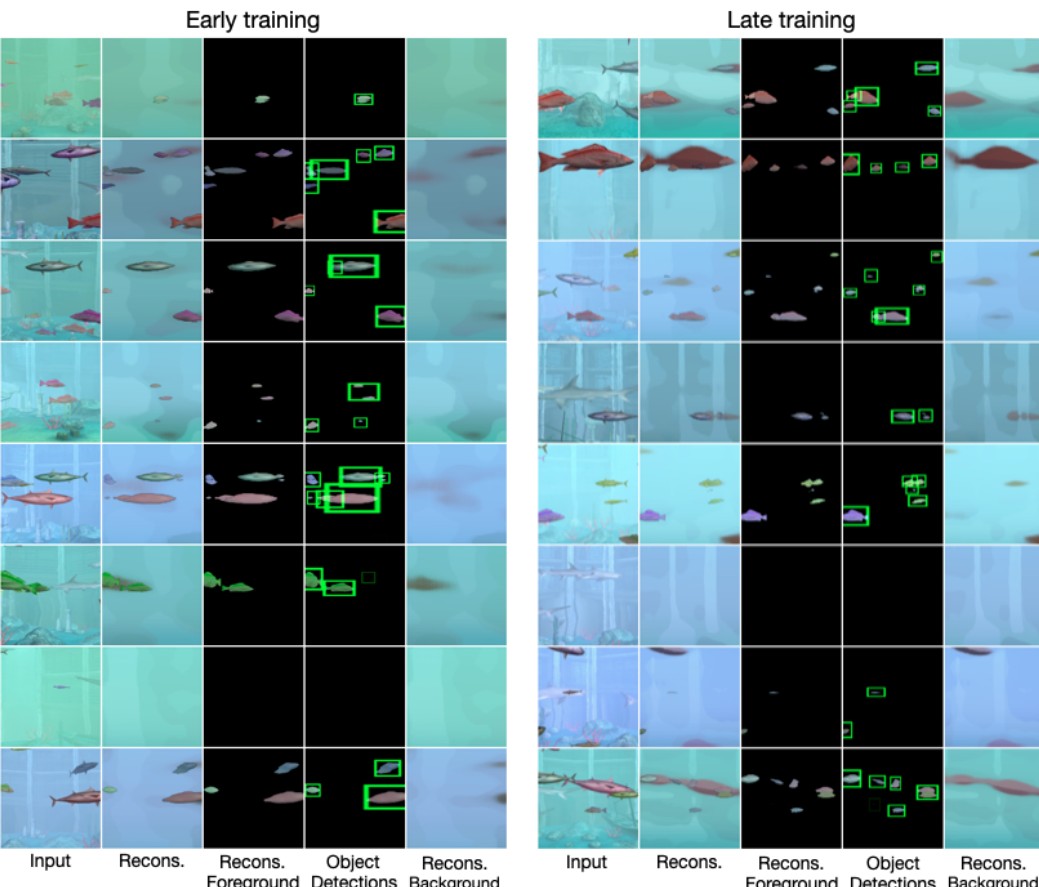

Figure 20: Scenes from the Fishbowl dataset reconstructed by SPACE (Lin et al., 2020b). The glimpse encoder and decoder where trained on the candidate objects given by the motion segmentation using the object model loss proposed in this work. Afterwards, the remaining components where trained end-to-end using the official implementation provided by the authors with the default parameters.

## D.3 GENESIS-V2 (ENGELCKE ET AL., 2021)

We trained GENESIS-v2 on the Fishbowl dataset using the official implementation by the authors (https://github.com/applied-ai-lab/genesis) with default hyperparameters except for the image resolution, number of object slots and batch size. We modified the model code to work for rectangular images and used a resolution of 128x192 pixels. We trained GENESIS-v2 having 5 object slots on 4 Nvidia RTX 2080Ti GPUs using a batch size of 64. Initial experiments with 10 object slots lead to the background being split up into multiple slots. As for the GAN baseline before, we used every 16th frame for training this model (160k frames in total).

In Fig. 21 we show qualitative results from GENESIS-v2 on the Fishbowl dataset. The reconstructions of the model look somewhat blurry but capture the major structure in the input images well.

Importantly, the visualization of the segmentation map and the individual slots reveal that the model succeeds in learning to decompose the scenes into background and objects. Sampling objects and scenes however fails with this model configuration. Most likely this happens due to the GECO objective (Rezende & Viola, 2018), that decreases the weight of the KL term in the loss function as long as the log likelihood of the input sample is below the target value. Training GENESIS-v2 with the original VAE objective instead of GECO leads to better scene samples, the decomposition of the scene however fails with this objective Fig. 22.

As a comparison to the end-to-end training within GENESIS-v2, we trained a variant of our object model using the architecture of the GENESIS-v2 component decoder. As encoder, we use a CNN constructed symmetrically to the decoder, using regular convolutions instead of the transposed convolutions. We trained the model with the Cutout augmentation and using the same loss and training schedule as for the object model described in the main paper.

The results in Tab. 5 and Fig. 23 show that the model generally performs well, but worse than our original object model. This can most likely be explained by the larger capacity of our object model (10 vs 5 layers, 128 vs 64 latent dimensions) which seems to be necessary to model the visually more complex objects in our setting. Also when trained separately, the object model has difficulties with sampling novel fish, which can be addressed at the cost of worse reconstructions 24.

Overall we conclude that our modular object learning approach scales much better to visually more complex scenes than end-to-end training as done by GENESIS-v2. Even when using the GENESIS-v2 component decoder within our framework, the necessary trade-off between reconstruction and generation capabilities seems to be more favorable when using our modular approach as opposed to the end-to-end training. We remark that this comparison comes with a grain of salt: We neither adapted the hyperparameters of GENESIS-v2 nor of our method and the same decoder is used within GENESIS-v2 for the background, too. The strong qualitative difference in sample quality however makes it unlikely that this explains all of the difference. Moreover, our approach only addresses learning generative object models whereas GENESIS-v2 is also capable to infer the decomposition of static input scenes. For the future, we therefore see much potential in combining the respective strengths of both methods.

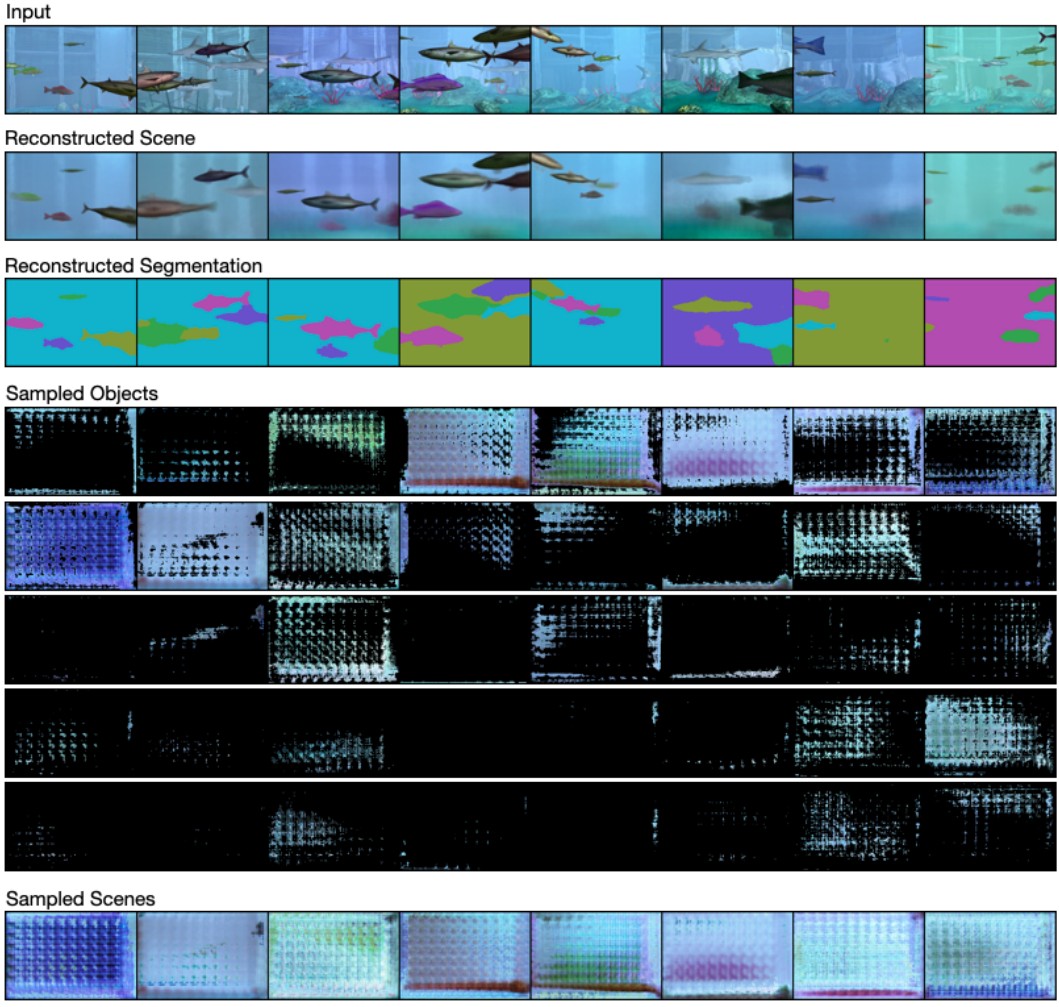

Figure 21: Qualitative results of GENESIS-v2 applied on the FISHBOWL dataset. The reconstruction in the second row look somewhat blurry, but capture all major structure in the input images shown in the first row. The visualization of the reconstructed segmentation shows that the model succeeds in decomposing the input image into the background and the different objects. Sampling from the model however fails in this setting.

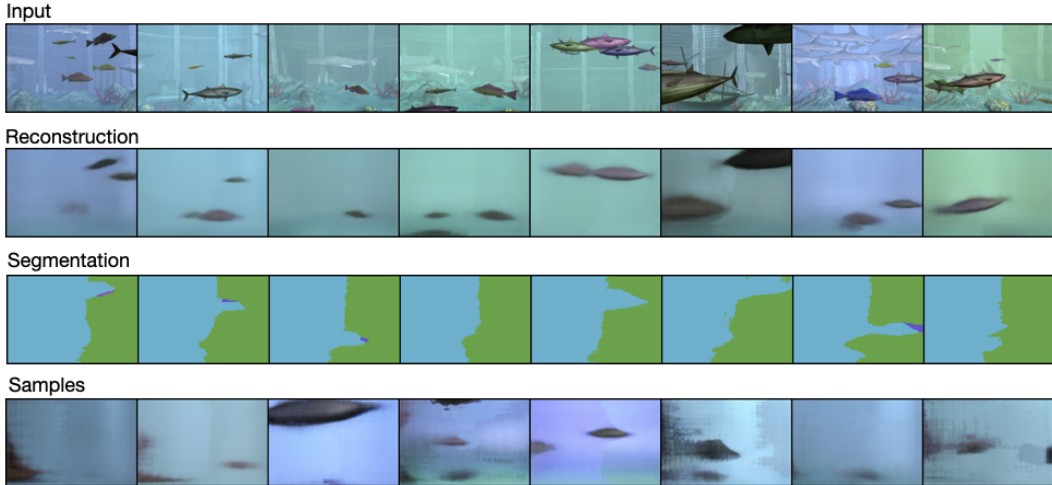

Figure 22: Qualitative results of GENESIS-v2 trained on the Fishbowl dataset using the default VAE objective instead of the GECO objective. Sampling from the model works much better in this setting; the model fails however to segment the scene into the indiviual objects.

Table 5: Comparison of the reconstructions from the object model based on the GENESIS-v2 component decoder with our original object model using the same metrics as in Tab. 2. Both models are trained using cutout augmentation.

| Training data | Architecture | IoU ↑ | MAE ↓ | IoU@0.5 ↑ | MAE@0.5 ↓ |
|---|---|---|---|---|---|
| Motion Segmentation | GENESIS-v2 | 0.779±0.002 | 15.3±0.063 | 0.661±0.003 | 25.7±0.114 |
| | Ours | 0.822±0.001 | 13.0±0.032 | 0.677±0.002 | 24.0±0.017 |
| Ground Truth Segmentation | GENESIS-v2 | 0.844±0.001 | 13.7±0.109 | 0.722±0.001 | 19.8±0.062 |
| | Ours | 0.887±0.001 | 12.3±0.035 | 0.743±0.003 | 17.3±0.087 |

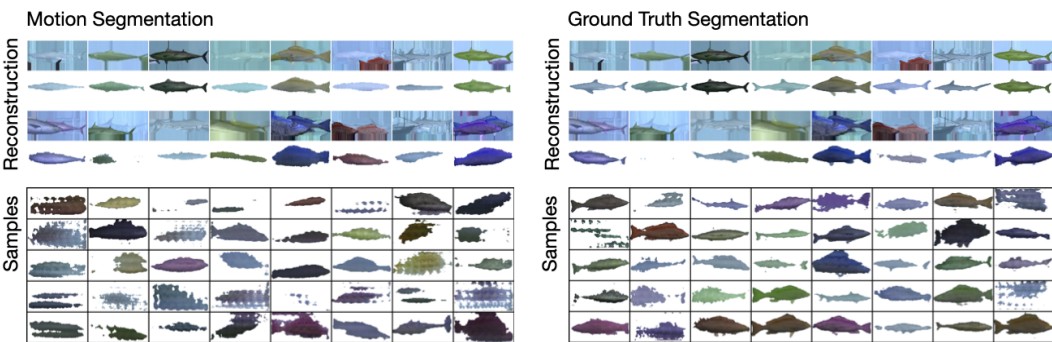

Figure 23: Qualitative results when using the GENESIS-v2 object decoder as object model within our modular training approach using the same loss and training schedule. Reconstructions and samples look worse than with the original object model, hinting at the larger capacity of our object model being necessary for our dataset.

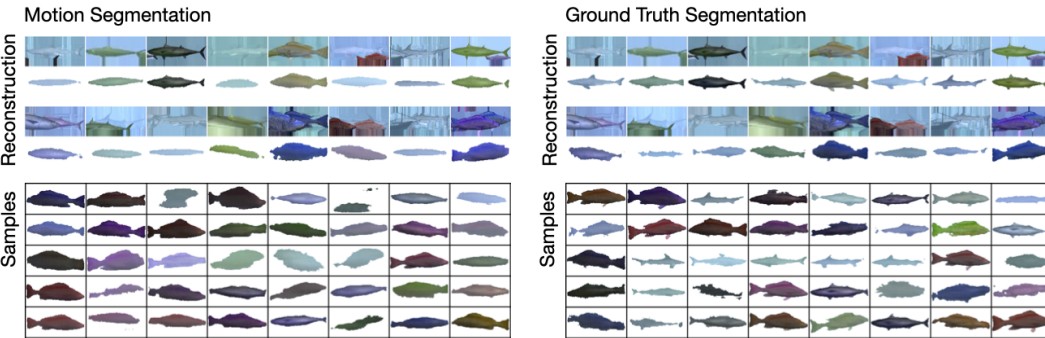

Figure 24: Qualitative results when using the GENESIS-v2 object decoder as object model within our modular training approach using a larger weight of the KL divergence in the VAE training loss. At the prize of worse reconstructions, the samples from the model can be substantially improved this way.

## D.4    COMPARISON ON THE REALTRAFFIC DATASET

To evaluate how well our method transfers to other settings, we trained our model on the RealTraffic dataset (Ehrhardt et al., 2020). As the resolution of the images is smaller in the RealTraffic dataset, we reduced the object size threshold to 64px and the minimal distance to the boundary to 8px for the object extraction. We trained the object model with the same architecture as used for the Fishbowl dataset. Due to the smaller dataset size, we trained the model for 600 epochs and reduced the learning rate only after 400 epochs. As the videos in the dataset were all recorded from the same stationary traffic camera, we did not train a background model but used the mean-filtered background directly. In contrast to the aquarium dataset, the object positions and sizes are not distributed uniformly for this dataset. For the scene model, we therefore sample directly from the empirical joint distribution of object positions and sizes (extracted from the object masks obtained by the motion segmentation).

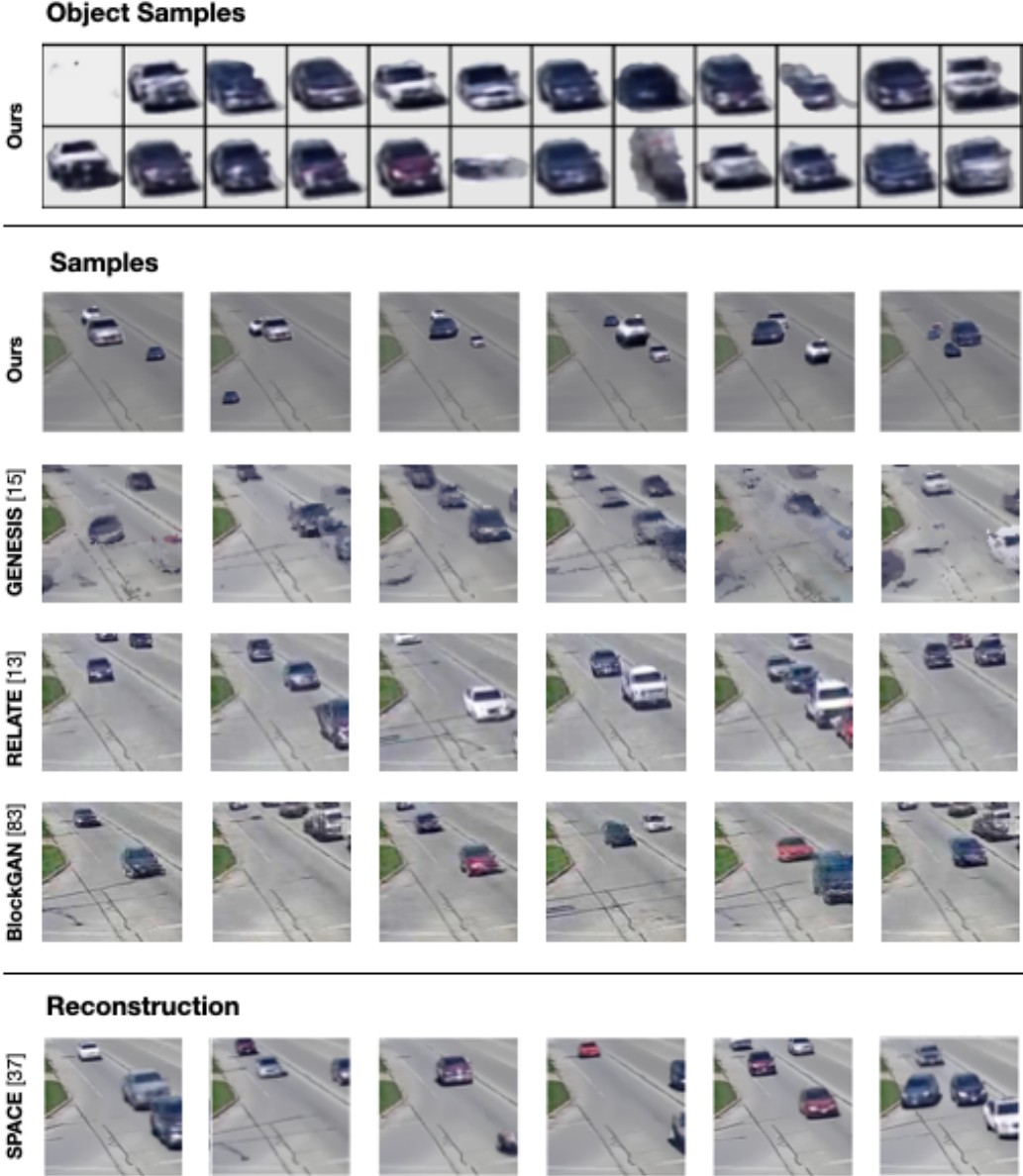

Figure 25: Results from our model trained on the RealTraffic dataset compared to results from other models trained on this dataset.

In Fig. 25, we compare samples from our model to other models for which results on the RealTraffic dataset have been reported in (Ehrhardt et al., 2020). Overall, our model transfers well to this real world setting, given that our model is used largely unchanged. In comparison to the GAN-based RELATE and BlockGAN, the samples from our model look slightly more blurry. However the results from our method clearly improve over the VAE-based GENESIS model.

The adaptation of the scene statistics as described above works well for the object position, as the cars are correctly positioned on the street only. We consider the possibility of this straight-forward adaptation to novel scene statistics to be a nice advantage of our object-centric modeling approach. In the future, this could be improved further by, e.g., learning to sample latent variables conditioned on the object positions.

