# OpenReview forum: "Unsupervised Object Learning via Common Fate"
_ICLR.cc/2022/Conference — ICLR 2022 Submitted_

### Official Review · Reviewer_dAqW · 2021-10-21

**Correctness:** 2
**Technical Novelty And Significance:** 1
**Empirical Novelty And Significance:** 2
**Recommendation:** 3
**Confidence:** 4

**Main Review:**

Object-centric generative models have gained relatively less attentions compared to the normal generation models. The authors have generated a new dataset for this research area. However, I have many concerns that make me hesitant to accept this paper.

FIrst of all, the dataset is far from the realistic setting. It is related to missing literature search. There are many datasets for amodal segmentation. For example, https://openaccess.thecvf.com/content_CVPR_2019/papers/Hu_SAIL-VOS_Semantic_Amodal_Instance_Level_Video_Object_Segmentation_-_A_CVPR_2019_paper.pdf <- this paper basically proposes a more realistic dataset, which provides the same information (amodal masks). I assume the authors have not looked into this literature. There are many other amodal datasets, including KINS and COCO-A.

My second point is that the proposed model has not been evaluated well. I have gone through the paper as well as the supplementary, but there was no appropriate comparison to the previous works. It's hard to believe all the conventional algorithms fail at generating object-centric images.

**Summary Of The Paper:**

This work proposes an object-centric generative model. It consists of motion segmentation, object model, background model, and scene model. The object model is trained to reconstruct an object as if it is not occluded. The authors have introduced a new dataset, called Fishbowl, which provides inmodal and amodal segmentation masks of objects.

**Summary Of The Review:**

Because of the above two major weaknesses, I'm skeptical of accepting this paper. I am open to change my rating depends on the rebuttal. For now, my initial rating is reject.

---

> ### Author Response · Authors · 2021-11-19
> **Re: Official Review of Paper1605 by Reviewer dAqW**
>
> We thank you for your review and would like to address the two major concerns you raised:
>
> 1. We agree that a clear gap exists between the Fishbowl dataset proposed in our work and natural videos. We appreciate the datasets you proposed and find them highly interesting. To our knowledge however, none of the existing object-centric methods has been shown to work on data of this complexity. Typically, very simple settings such as moving MNIST digits or simple geometric shapes have been used by previous works. Therefore we think that our dataset which is positioned between those simple toy settings and more realistic settings as proposed in the papers you mention is a meaningful contribution. Given the difficulties of scaling object-centric methods, we see our dataset as an important milestone for scaling object-centric representation learning to real world data.
>
> 2. The setting tackled in our work indeed lacks substantially behind what is possible in other settings (e.g. supervised), however we’re not aware of any object-centric models that have been shown to scale to the complexity of the Fishbowl dataset. Moreover, as detailed in our response to reviewer GUfQ we only compared qualitatively to previous works since we're not aware of a principled method to compare generative models. Nevertheless we’re open to suggestions which additional models and comparisons would be interesting and happy to include them in our paper to strengthen the analysis of our method.

---

> > ### Comment · Reviewer_dAqW · 2021-11-22
> > **Following concerns**
> >
> > Thank you for your comments. I still have a question regarding the literature review though. The amodal datasets that I referred are not like moving MNIST digits or simple geometric shapes. They are composed of realistic or synthetic videos with high complexity. Is there any reason that they are not usable for this task?

---

> > > ### Author Response · Authors · 2021-11-23
> > > **Re: Following concerns**
> > >
> > > We don’t see any reason that those datasets are not usable for this *task*. Rather we believe that they are not usable for the current *models*. As you stated they “are not like moving MNIST digits or simple geometric shapes”, but “composed of realistic or synthetic videos with high complexity”. Previous models have not yet been shown to work on datasets of this complexity. As shown in our work, even the simpler Fishbowl dataset is difficult for existing methods.

---

### Official Review · Reviewer_3iwv · 2021-11-01

**Correctness:** 3
**Technical Novelty And Significance:** 2
**Empirical Novelty And Significance:** 3
**Recommendation:** 6
**Confidence:** 3

**Main Review:**

I found the paper quite useful and interesting. I find it useful because the decomposition allows for better ablative analysis of the model performance compared to previous end-to-end approaches, and because it opens up additional practical applications, such as given user direct and interpretable control on the output generated by the model (e.g. manually input number of objects or an individual object position). I find the results interesting, particularly the thorough ablative analysis: comparing training on ground truth segmentation vs training on motion segmentation.

The paper has some weaknesses, e..g. some claims not thoroughly backed up, lack of clarity on some details. I list them here in detail, from more important to less important:
1) The paper strongly couples two concepts: a) the choice of using a multi-stage learning approach rather than end-to-end; and b) the use of motion segmentation to enable training this model. In practice, it seems to me that these two things are not really that coupled, but rather motion segmentation is just a way to automatically generate the training data that the model needs, because unlike end-to-end method, this submission needs explicit object and segmentation masks (if I understand correctly). One thing I would like the authors to highlight more  is that, due to this, a limitation of this method is that it can be trained only on video data and not on static images, unless some form of segmentation ground truth is available or there is a way to automatically generate it -  this is mentioned in the conclusions, but I encourage making it clear earlier in the paper
2) It is unclear if the parameters of the scene models learnt entirely independently from the foreground and background model, i.e. by learning the distribution of the latent vectors produced by the VAEs conditioned on the background? Or is there some gradient propagating between these steps
3) von kugelgen et al (2020) also provide a controllable way to change the way the model generates the scenes (e.g. number and type of objects). Can you highlight the main differences? Is it mostly that in this submissions there is also control on scale and position?
4) Results on the fishbowl dataset look quite impressive, less on the car dataset (I suggest adding some visuals from this dataset and of visual comparisons to existing methods in the main body of the paper). It would be interesting to see this method run on other datasets like the Atari dataset or the 3D room dataset from the SPACE paper, to further test generalisation of the multi-stage approach in other domains - do ground truth segmentations exist to test there?
5) Understanding the performance of this methods on generating new scenes compared to existing work is quite challenging because it is done mostly qualitatively, although I recognise that is a challenge of the domain and existing methods also seems to suffer from this

**Summary Of The Paper:**

The paper introduces an object-centric generative model for visual scenes. The model decouples the problem into three tasks: 1) modelling the  2D appearance and shape of individual objects with a variational auto encoder; 2) same thing for background; 3) sampling the position, size and appearance of individual objects (i.e. scene composition) conditioned on the background. To my understanding, these three components are trained independently.

Contributions:
1) Decoupling these three tasks allows for a more interpretable representation compared to existing end-to-end methods, which allows for better interactions with users (e.g. change the number of objects or their position)
2) This decoupling potentially makes the third task easier, because it only needs to learn relationships between positions and size given a latent representation (as opposed to learning everything jointly).
3) Training the tasks independently requires a way to obtain "ground-truth" object segmentations - the paper achieves this automatically from videos using motion segmentation with good results

**Summary Of The Review:**

While I do not see a significant amount of novel technical contributions, the proposed paradigm of breaking down scene generative models into multiple stages of learning that allows encoding prior structure and human-interpretable elements is valuable. It can help generate theoretical insights, and the ablative analysis provided here is a promising first step. It also has the potential to be practically useful, by allowing users more fine-grained controls on the parameters of the generated scenes.

There are some claims with minor issues, and some lack of clarity in some sections (see weaknesses in the review section above). All in all, I think the paper is above the acceptance threshold, and addressing these weaknesses would make me lean towards a clearer accept

---

> ### Author Response · Authors · 2021-11-19
> **Re: Official Review of Paper1605 by Reviewer 3iwv**
>
> Thank you for your review! We appreciate that you find our multi-stage approach “valuable”, having the “potential to be practically useful” and that you “think the paper is above the acceptance threshold”. In the following we would like to address your concerns.
>
> We agree that, in principle, using motion segmentation is independent from the multi-stage approach. However we think that using motion segmentation is a natural choice to obtain unsupervised segmentation information. The common fate Gestalt principle is a strong cue for object boundaries that is also assumed to play a central role for object learning in humans (Spelke 1990). We’re not aware of purely image based heuristics being equally reliable. Naturally our approach requires temporal data for training, we however don’t expect this to be a major limitation in practice since videos are usually easy to obtain and our approach doesn’t require any ground truth. We’re happy to discuss this earlier in our paper.
>
> Furthermore, we would like to clarify the aspects you mentioned:
>
> 2. The parameters of the scene model are indeed independent from the object and background models. For sampling objects and backgrounds, we sample latent vectors from the respective prior distributions and use the scene model only to compose the objects and background into a scene. A tighter coupling would be possible (e.g. by using the scene model to generate latent object or background vectors instead of the priors), we however left exploring this for future work.
> 3. Beyond the more fine-grained control, we see the main difference in the scalability of the method. While the end-to-end approach in von Kügelgen et al. 2020 was limited to very simple settings, the combination with motion segmentation as done in our work allows scaling to more realistic settings.
> 4. We chose to include the results on the RealTraffic dataset in the appendix to obey the page limit, and found the present distribution of content to be adequate. However, we're open to suggestions on what to move to the appendix when including the results on the RealTraffic dataset in the main paper.
> 5.The Object Rooms dataset does not contain videos, so as discussed above we cannot apply our method to it. It would be possible to apply our method to the Atari dataset in principle, however some objects in this dataset do not undergo independent motion, requiring additional segmentation heuristics beyond the motion segmentation.
> 6. We agree, comparing samples from generative models is a major challenge and we’re not aware of any principled quantitative comparison (cf. Theis et al. 2016, https://arxiv.org/pdf/1511.01844.pdf). As often done, we therefore only compare qualitatively.

---

> > ### Comment · Reviewer_3iwv · 2021-11-21
> > **Question on Objects Room dataset**
> >
> > Could you further clarify why it is not possible to evaluate on ObjectRooms? I appreciate that this dataset is not videos, but I thought that the method in this submission would work on any object masks (I am assuming that they do not come from motion). Is it because ObjectRooms does not provide object masks (since the dataset is rendered I assumed they were available).

---

> > > ### Author Response · Authors · 2021-11-23
> > > **Re: Question on Objects Room dataset**
> > >
> > > Yes, it would be possible to run our method using the ground truth masks provided by the dataset. However we’re not sure how much additional information we would obtain by this experiment on it’s own. In our paper, we’re proposing an unsupervised object learning approach, which we cannot apply on the ObjectRooms dataset due to the missing temporal information. We only use the ground truth information for estimating how much our object model could be improved by improving the motion segmentation.
> > >
> > > In response to reviewer GUfQ we followed an alternative approach for increasing the comparability of our method to SPACE: We pretrained the object model included in SPACE using our approach, and then trained the remaining components end-to-end. This experiment qualitatively and quantitatively showed that our approach allows us to improve over the original, end-to-end SPACE. The off-the-shelf SPACE however still falls short of the scene model proposed in our work, even when pretraining the object model.

---

### Official Review · Reviewer_GUfQ · 2021-11-02

**Correctness:** 2
**Technical Novelty And Significance:** 3
**Empirical Novelty And Significance:** 3
**Recommendation:** 3
**Confidence:** 4

**Main Review:**

Strengths:
1. I appreciate the effort of building a new dataset.
2. Compared with other methods, the scalability of the proposed method is indeed strong.
3. I am convinced that the direct application of "the common fate heuristic" is novel and is of great potential.
4. While most related work can only generate masks for occluded (partial) objects, the proposed method can generate masks for full objects occluded or not.
5. The ability to sample novel scenes is impressive.

Weakness:
I apologize if some of my concerns are already detailed in the paper and I fail to pick them out.
1. Discussion of related works.
    While the author did a comprehensive literature review, I am not convinced by some of the claims.
    1): I am not sure why "the spatial mixture model does not match the true, underlying scene generation process".
         For example, SPACE employs a depth prediction to predict the relative depth between objects.
         Then the depth information is integrated into the mask value.
         SPACE effectively transforms a 2D segmentation task into 2.5D.
    2): "End-to-End object-centric approaches are difficult to train."
         Without further explanation, I am not sure what "difficult to train" means under the context.
    3): "Full scene models trained in an end-to-end fashion are not shown to scale to a more realistic dataset yet."
         As demonstrated by the paper SCALOR: GENERATIVE WORLD MODELS WITH SCALABLE OBJECT REPRESENTATIONS (which is already cited by the author), such method can be applied to challenging cases.

2. Dataset
The dataset proposed is indeed interesting and challenging.
My only question is that is it true that all foreground objects are moving all the time and the backgrounds do not have any moving parts?

3. Baseline
It is surprising that no baseline model is evaluated on the same dataset.
I understand that there are not many works targeting exactly the same setup, but works like SCALOR or SPACE can also be used as a baseline for numerical results.
After all, being able to segment without "the common fate" heuristic and a rule based object proposal stage seems like an advanrages to me.
To show that the loss function proposed in this work is better, the author can modify SPACE to take bounding boxes proposed by the rule based motion segmentation pipeline. This should be an easy modification.
If under this modification the performance of SPACE is better than before but still behind this paper, then the method propsed in this work is truely great.
Similar modification can be done to SCALOR as well I think.


**Summary Of The Paper:**

This paper provides a multi-stage solution to unsupervised, frame-wise segmentation in videos.
The common fate heuristic is used to provide initial object detections and segmentations.
Then VAE based model is used to refine the initial results.
A new simulated dataset is proposed.

Results show that the proposed method successfully segments out moving objects and is highly scalable.

**Summary Of The Review:**

The results provided in the paper is impressive.
But the authors argue multiple times that the method proposed in this work is better than end-to-end generative model without enough justifiation.
Thus I cannot say that this paper is above accept threshold at current stage.

---

> ### Author Response · Authors · 2021-11-19
> **Re: Official Review of Paper1605 by Reviewer GUfQ**
>
> We thank you for your comments and appreciate that you find ‘the scalability of the proposed method [...] indeed strong’ and that ‘"the common fate heuristic" is novel and [...] of great potential’. In the following, we address your concerns in greater detail.
>
> You found it “surprising that no baseline model is evaluated on the same dataset.” We trained SPACE and GENESIS-v2 on the Fishbowl dataset and provide a detailed discussion of the results in the supplement. We however didn’t compare quantitatively for two reasons:
>
> - Previous works often focussed on evaluating the segmentation. With our approach, the segmentation is handled by the motion segmentation so that we would not compare to our actual contribution this way.
> - Quantitatively evaluating samples from generative models is challenging and we’re not aware of a principled method for this comparison (cf. Theis et al. 2016, https://arxiv.org/pdf/1511.01844.pdf).
>
> A qualitative difference to SPACE is that in our approach the scene composition is based on a dead leaves model. SPACE reconstructs a mean image based on a weighted sum of all components, with the depth influencing the weight. This means that a pixel value reflects a linear interpolation between objects (e.g., 30%-70%), which is physically implausible (unless for translucent objects, which are not part of the datasets considered). In contrast, in our model every pixel corresponds to 100% to the most foreground object. This approach allows us to freely add or move objects in a scene without having to adapt the mixture weights for the other objects, making the object representations truly independent (as demonstrated in Fig. 6) and closer to the real generative model.
>
> In the dataset we introduced, all foreground objects (fish) are indeed moving at various speeds. The background is mainly static, except for small periodic movements of, e.g., plants. We agree that visually more complex datasets have been used by some methods and highly appreciate the results of, e.g., SCALOR in that direction. In our work however, we primarily aim at learning visually more complex *objects*, whereas the complexity addressed by SCALOR is the complexity of the overall scenes (i.e., the object count). Moreover, despite some methods showing results on visually more challenging datasets, the quantitative evaluation is typically not feasible in this setting since ground truth information is missing. This is the primary gap closed by our novel dataset.
>
> As the main ingredient that allows our model to scale to this relatively complex setting, we see the decoupling of the object detection and generative object modelling that is enabled by the motion segmentation. We argue that this simplifies training regarding two aspects:
>
> - Detecting objects and finding a suitable latent object representation are problems that mutually depend on each other. End-to-end models aim at solving both problems at once. Our approach solves these subproblems sequentially, which can be seen as a successful curriculum learning strategy.
> - As appreciated by reviewer 3iwv, this separation into individual stages allows testing and optimizing each stage individually. For example, testing a different architecture for the object decoder only requires retraining the object model, which is relatively fast (<3h on a single GPU). An end-to-end approach on the other hand requires training the entire model from scratch, which is computationally much more expensive.
>
> We primarily see our work as a proof-of-concept that the motion segmentation based decoupling is a successful approach for scaling object-centric modeling. Beyond the evidence provided in our paper, we think your suggestion to integrate our approach with SPACE is an interesting experiment and agree that “If under this modification the performance of SPACE is better than before but still behind this paper, then the method proposed in this work is truely great.” We used the bounding boxes from the motion segmentation to pretrain the glimpse encoder and decoder from SPACE, followed by end-to-end training of the entire model. Early in training, this approach largely solves the discussed failure mode of objects being split across several bounding boxes, but eventually leads to many objects being modelled by the background component. We included this experiment with a Figure showing the results in the supplement of our paper (figure link: https://ibb.co/yFtsnHY).
>
> While preliminary, those results demonstrate that our approach can also be used to improve existing object-centric models, but also show that the off-the-shelf SPACE model trained this way still falls short of the scene model proposed in our work. Nevertheless, the results in this setting look promising and we expect a better integration of our approach with state-of-the-art object centric architectures to eventually surpass both SPACE and our scene model.

---

> > ### Comment · Reviewer_GUfQ · 2021-11-21
> > **Some other concerns**
> >
> > 1. I appreciate that the authors show new experiments in such a short time frame.
> > The qualitative results indeed show that SPACE equipped with motion segmentation does not perform as well in the late training.
> > Did the author tune the hyperparameters of the SPACE to better suit the current dataset?
> > Can the author share some insights about why SPACE tends to over-segment in the end?
> > Particularly, when the segmentation results of SPACE degenerate, does the overall loss still decrease?
> >
> > 2. Regarding the stage-wise design, I am hardly convinced that it can be classified as curriculum learning.
> > In stage one, there is no "learning".
> > In fact, I may describe the first stage as data preprocessing.
> > That is to say, the deep learning model proposed in this paper is end-to-end but requires a rough object bounding box to start with.
> > I am not saying that the stage-wise pipeline is a shortcoming, but I do not consider it as an advantage.
> > I'd rather consider it as a small and fair enough price to pay for using the dead leaves model.
> > I know it is somehow irresponsible to comment by feelings but I feel that the dead leaves model only changes the differentiability of the masks and thus does not necessarily rely on any rule-based mask proposal to work.
> >
> > 3. I would not ask the question below if the proposed model is end-to-end.
> > But since it contains rule-based parts, I have to ask how much of the problem can be solved within stage one with currently available techniques.
> > Under the assumption that for every fish there exists at least one frame where the fish is not blocked by any other fish, is it possible to obtain perfect segmentation with a rule-based method?
> > Then a simple VAE can be trained and this will make the entire training process even easier.
> >
> > 4. I firmly believe that quantitative result is a must-have.
> > Please allow me to ask this one question: Without any quantitative results, how another work can demonstrate its superiority over this work? Comparing qualitative results again?
> > Then all benchmarks we have on this dataset are a set of images.
> > I do not believe that is healthy for the community.
> > Since the dataset is proposed in this work, then a set of well-defined metrics must be found also in this work to accompany the dataset.

---

> > > ### Author Response · Authors · 2021-11-23
> > > **Re: Some other concerns**
> > >
> > > Thank you for your feedback on our response and appreciating the new experiment.
> > >
> > > As stated in our paper, we agree that a complete end-to-end approach would be more elegant. With advances in building structured generative models it is very well conceivable that an approach building on a dead-leaves model might be trained with end-to-end methods eventually. End-to-end approaches however have not been shown to scale to more realistic settings yet (as we also confirm for multiple models on our dataset), while the multi-stage approach used in our setting does. We agree that the term curriculum learning might be misleading in this context and will avoid it in our work.
> > >
> > > The usefulness of our multi-stage approach is further reinforced by the fact that it readily helps to scale end-to-end models such as SPACE. Beyond the qualitative results, quantitatively evaluating the object model of SPACE confirms this point (see below). We could not afford to run a complete hyperparameter search, so it is possible that the model can be improved further. We however see this experiment mainly as a proof-of-concept that our method is flexible enough so that it can be readily integrated with existing end-to-end architectures, and leave improving the integration for future work. We’re not certain what causes the difficulties of SPACE in this setting. We can confirm that the overall loss is decreasing during the training (Figure: https://ibb.co/V2dQhQJ), so the loss used by SPACE favors an unintended solution in this setting.
> > >
> > > We can well understand your concerns regarding the qualitative evaluation. However, this is not a shortcoming of the specific dataset we are proposing. Other datasets commonly used for object-centric modeling (e.g. https://github.com/deepmind/multi_object_datasets) do not provide metrics for evaluating generative modeling either. The problem persists for generative modeling as such: There is no accepted principled method for evaluating samples from generative models, and we think being honest about this shortcoming is preferable over focussing on a suboptimal performance metric. We see improving this situation beyond the scope of our work.
> > >
> > > Moreover, the qualitative analysis only affects the samples from the overall scene model. We respectfully disagree that our work comes “without any quantitative results”, since we’re quantitatively evaluating the capabilities of our object model (Table 2). We acknowledge that evaluating SPACE using this approach improves the comparability and added this information to our submission. As the quantitative results in Table 4 (direct link: https://ibb.co/vPRD9NR) show, the reconstruction of the appearance works better with the SPACE object model than with our approach*, but has a substantial gap in the segmentation performance. Most importantly, the results clearly show an improvement of the motion segmentation based training over the original end-to-end approach, confirming the qualitative results. A future method for training object models “can demonstrate its superiority over this work” by showing a quantitative improvement of the learned object model using those metrics.
> > >
> > > *) To some degree this is confounded with the worse segmentation performance, as the appearance error is only evaluated on the intersection of the ground truth and predicted mask.

---

> > > > ### Comment · Reviewer_GUfQ · 2021-11-30
> > > > **Reply to the authors's rebuttal**
> > > >
> > > > Given the author's rebuttal, I would like to keep my rating.
> > > >
> > > > While the idea is novel and the results look good, the comparison with the baseline needs to be more thorough.
> > > > The hyperparameter needs to be carefully tuned.
> > > > For example, the glimpse shape prior in SPACE should reflect the fact that fishes tend to be thin and long.
> > > > Image grid cell size should also be tuned carefully.
> > > > All the hyperparameters of the baseline should be also presented in the paper for the baseline to be convincing.

---

### Author Response · Authors · 2021-11-23
**Meta Response**

We would like to thank all reviewers for their feedback and valuable results. We’re happy to hear that you find “the scalability of the proposed method [...] indeed strong” (GUfQ), “useful and interesting” (3iwv). The discussion has led to some clarifications and a new, interesting experiment. In the following we would like to summarize those updates.

We mainly see our method as a proof of concept that unsupervised motion segmentation allows scaling object centric generative modelling to visually more complex settings. While it is conceivable that end-to-end training of object-centric models eventually succeeds also for more realistic inputs, our approach based on off-the-shelf unsupervised motion segmentation already succeeds now.

A major point raised by several reviewers concerns the comparison to previous works in the field. Beyond training existing methods on our data we performed an additional experiment in which we integrated our approach with the end-to-end SPACE model: We first pretrained the object encoder and decoder of SPACE based on motion segmentations and using the same loss as for our object model. We then inserted this pretrained object model into the full SPACE model and trained the remaining components end-to-end. The quantitative and qualitative evaluation revealed that this training scheme clearly outperforms the end-to-end trained SPACE object model, which demonstrates that our approach can be readily integrated with existing architectures. However, those experiments also revealed that the off-the-shelf SPACE model still falls short of our scene model even when pretrained. In our view, these results contribute clear evidence for the usefulness of our approach and we see improving the integration with end-to-end architecture as a promising path towards applying object-centric modeling in complex, real world settings.

---

### Decision · Program_Chairs · 2022-01-20

**Decision:**

Reject

**Comment:**

This paper studies the challenging problem of object-centric generation of visual scenes. While the paper has some novel ideas that make it interesting, its (quantitative and qualitative) comparison with existing methods is currently premature to allow drawing conclusions with sufficient evidence.

Instead of claiming that existing models cannot do well for the more realistic datasets mentioned by reviewer dAqW, it would be more convincing to conduct a comprehensive experimental study by comparing the proposed method with existing methods on a range of datasets, from simple ones to more realistic ones. The synthetic Fishbowl dataset introduced in this paper can be one of them.

Moreover, the clarity of the paper could be improved to make it appeal better to the readers.

All three reviewers engaged actively in discussions (both including and not including the authors). Although one reviewer recommends 6 (weak accept), the reviewer also shares some of the concerns of the other reviewers. As it stands, the paper is not ready for acceptance. If the comments and suggestions are incorporated to revise the paper, it will have potential to be a good paper for future submission.